# Tight High Probability Bounds for Linear Stochastic Approximation with Fixed Stepsize

**Alain Durmus**
Université Paris-Saclay, ENS Paris-Saclay, CNRS,
Centre Borelli, F-91190 Gif-sur-Yvette, France `alain.durmus@ens-paris-saclay.fr`

**Eric Moulines**
Ecole Polytechnique and HSE University
`eric.moulines@polytechnique.edu`

**Alexey Naumov**
HSE University
`anaumov@hse.ru`

**Sergey Samsonov**
HSE University
`svsamsonov@hse.ru`

**Kevin Scaman**
INRIA and ENS, PSL University
`kevin.scaman@inria.fr`

**Hoi-To Wai**
The Chinese University of Hong Kong
`htwai@se.cuhk.edu.hk`

## Abstract

This paper provides a non-asymptotic analysis of linear stochastic approximation (LSA) algorithms with fixed stepsize. This family of methods arises in many machine learning tasks and is used to obtain approximate solutions of a linear system $\bar{A}\theta = \bar{b}$ for which $\bar{A}$ and $\bar{b}$ can only be accessed through random estimates $\{(\mathbf{A}_n, \mathbf{b}_n) : n \in \mathbb{N}^*\}$. Our analysis is based on new results regarding moments and high probability bounds for products of matrices which are shown to be tight. We derive high probability bounds on the performance of LSA under weaker conditions on the sequence $\{(\mathbf{A}_n, \mathbf{b}_n) : n \in \mathbb{N}^*\}$ than previous works. However, in contrast, we establish polynomial concentration bounds with order depending on the stepsize. We show that our conclusions cannot be improved without additional assumptions on the sequence of random matrices $\{\mathbf{A}_n : n \in \mathbb{N}^*\}$, and in particular that no Gaussian or exponential high probability bounds can hold. Finally, we pay a particular attention to establishing bounds with sharp order with respect to the number of iterations and the stepsize and whose leading terms contain the covariance matrices appearing in the central limit theorems.

## 1 Introduction

This paper provides a detailed analysis of Linear Stochastic Approximation (LSA) schemes which aim at finding a solution $\theta^\star$ for a linear system of the form $\bar{A}\theta = \bar{b}$. In particular, we analyze LSA with a fixed stepsize $\alpha > 0$ which consists in defining a sequence of estimates $\{\theta_n : n \in \mathbb{N}\}$ for $\theta^\star$ by the recursion

$$\theta_{n+1} = \theta_n - \alpha\{\mathbf{A}_{n+1}\theta_n - \mathbf{b}_{n+1}\}, \quad n \in \mathbb{N}, \tag{1}$$

where $\{(\mathbf{A}_n, \mathbf{b}_n) : n \in \mathbb{N}^*\}$ is a sequence of i.i.d. random variables used as proxy for $\bar{A} \in \mathbb{R}^{d \times d}$ and $\bar{b} \in \mathbb{R}^d$ which are typically unknown. This class of algorithms and the corresponding setting have a long history and important applications in signal processing such as channel equalization and echo cancellation [3, 23]. It has renewed interests in machine learning and computational statistics especially for least-square estimation and Reinforcement learning (RL) [4, 7, 37]. The recursion (1) has already been studied in depth in several works which derive asymptotic [33, 23, 6, 3] and non-asymptotic [35, 27, 2, 20, 21, 5, 24, 36, 10, 13] guarantees.

35th Conference on Neural Information Processing Systems (NeurIPS 2021).

However, in most cases, there is a consistent gap between these two types of analyses. While asymptotic analysis gives important insights on the qualitative convergence of (1) based on statistical key quantities of the problem on hand, they do not provide finite-time convergence, or high probability bounds, necessary to obtain non-asymptotic confidence sets, see [28, 11] and the references therein. On the other hand, non-asymptotic studies are in general too coarse and lose significant statistical information in their derivation. Further, their upper bounds are generally loose when used in predicting the actual performance of LSA. We aim at filling this gap and provide conditions on $\{(\mathbf{A}_n, \mathbf{b}_n) : n \in \mathbb{N}^*\}$ ensuring tight high probability bounds on the sequence $\{\theta_n : n \in \mathbb{N}\}$.

This problem has been addressed in several contributions but at the expense of strong conditions on the sequence $\{(\mathbf{A}_n, \mathbf{b}_n) : n \in \mathbb{N}^*\}$. [14] provided concentration bounds for non-linear stochastic algorithms under a log-Sobolev condition which turns out to be hard to verify for most applications except for the Euler-Maruyama discretization scheme applied to Stochastic Differential Equation. [29] derived concentration inequalities but assuming that the innovations in (1) are uniformly bounded. Another popular, yet restrictive condition (see [25]) is that for any $n \in \mathbb{N}$ and $\theta \in \mathbb{R}^d$, the matrix-vector products $(\mathbf{A}_n - \bar{A})\theta$ are sub-Gaussian with parameter not depending on $n$ and $\theta$. In contrast, we aim at giving simple and mild conditions ensuring high probability bounds. More precisely, one of our key contributions (Theorem 1) is to show that under mild conditions on the sequence $\{(\mathbf{A}_n, \mathbf{b}_n) : n \in \mathbb{N}^*\}$, for any $\delta \in (0, 1)$, $n \in \mathbb{N}$ and $u \in \mathbb{S}^{d-1}$,

$$
\mathbb{P}\left( \left| u^\top(\theta_n - \theta^\star) \right| \le c\sqrt{\alpha}\left( \sqrt{u^\top \mathbf{\Sigma} u} + \sqrt{\alpha} \right) \sqrt{\log\left(\frac{1}{\delta}\right)} + c\,\frac{\rho_\alpha^n + \alpha\, p_0^2}{\delta^{\frac{1}{p_0}}} \right) \ge 1 - \delta\,, \quad (2)
$$

where $\rho_\alpha \in (0, 1)$, $c > 0$ is a constant independent of $n, \alpha, \delta$, and $p_0 = o(\alpha^{-1/4})$. In the above, $\mathbf{\Sigma}$ is the unique solution of the Lyapunov equation which naturally appears in central limit theorems for LSA with diminishing stepsize [cf. (26)]. In addition, we show that the bound we get is tight with respect to $\alpha$ and $\delta$ in the case where we only assume that $-\mathbb{E}[\mathbf{A}_1] = -\bar{A}$ is Hurwitz. Indeed, we provide counterexamples illustrating that for a fixed stepsize $\alpha$ and under the conditions that we consider, logarithmic dependence in $1/\delta$ cannot hold in (2) but only a polynomial one. Regarding the dependence with respect to $\alpha$, we extend [30] and show that for $\alpha$ small enough, $\{\theta_n : n \in \mathbb{N}\}$ admits a unique stationary distribution $\pi_\alpha$ and establish a central limit theorem for this family of distribution as $\alpha \downarrow 0$ at rate $\sqrt{\alpha}$ and with asymptotic covariance matrix $\mathbf{\Sigma}$ appearing in (2).

Finally, our proofs rely on a new analysis of product of matrices which extends the recent work in [18]. In particular, we establish conditions ensuring uniform bounds in $n$ of the $p$-th moments of $\mathbf{Y}_n \cdots \mathbf{Y}_1$, where $\{\mathbf{Y}_n : n \in \mathbb{N}^*\}$ is a sequence of independent matrices whose expected values have a spectral radius strictly less than 1. In comparison to existing results, the main challenge addressed is that the random matrices $\{\mathbf{Y}_n : n \in \mathbb{N}^*\}$ are not required to be almost surely symmetric.

The paper is organized as follows. Section 2 formally discusses the assumptions on LSA for our analysis. Section 3 presents the moment bound for product of random matrices. Using this result, Section 4 shows the high probability concentration inequality (2) and Section 5 shows the tightness of the bounds by deriving a central limit theorem for LSA.

**Notations**   Denote $\mathbb{N}^* = \mathbb{N} \setminus \{0\}$ and $\mathbb{N}_- = \mathbb{Z} \setminus \mathbb{N}^*$. Let $d \in \mathbb{N}^*$ and $Q$ be a symmetric positive definite $d \times d$ matrix. For $x \in \mathbb{R}^d$, we denote $\|x\|_Q = \{x^\top Q x\}^{1/2}$. For brevity, we set $\|x\| = \|x\|_{\mathrm{I}_d}$. We denote $\|A\|_Q = \max_{\|x\|_Q=1} \|Ax\|_Q$, and the subscriptless norm $\|A\| = \|A\|_\mathrm{I}$ is the standard spectral norm. We denote the condition number of $Q$ as $\kappa_Q = \lambda_{\min}^{-1}(Q)\lambda_{\max}(Q)$. For $B \in \mathbb{R}^{d \times d}$, we denote by $(\sigma_\ell(B))_{\ell=1}^d$ its singular values. For $p \ge 1$, the Schatten $p$-norm is denoted by $\|B\|_p = \{\sum_{\ell=1}^d \sigma_\ell^p(B)\}^{1/p}$. For $p, q \ge 1$ and random matrix $\mathbf{X}$, we write $\|\mathbf{X}\|_{p,q} = \{\mathbb{E}[\|\mathbf{X}\|_p^q]\}^{1/q}$.

We denote $\mathbb{S}^{d-1} = \{x \in \mathbb{R}^d | \|x\| = 1\}$. Let $A_1, \ldots, A_N$ be $d$-dimensional matrices. We denote $\prod_{\ell=i}^j A_\ell = A_j \ldots A_i$ if $i \le j$ and with the convention $\prod_{\ell=i}^j A_\ell = \mathrm{I}_d$ if $i > j$. We say that a centered random variable (r.v.) $X$ is sub-Gaussian with variance factor $\sigma^2$ and we denote $X \in \mathrm{SG}(\sigma^2)$ if for all $\lambda \in \mathbb{R}$, $\log \mathbb{E}[e^{\lambda X}] \le \lambda^2 \sigma^2 / 2$. We define the Wasserstein distance of order 2 between two probability measure $\mu$ and $\nu$ on $\mathbb{R}^d$ as $W_2(\mu, \nu) = \inf_{\zeta \in \Pi(\mu,\nu)} \int_{\mathbb{R}^{2d}} \|x - y\|^2 \mathrm{d}\zeta(x, y)$, where $\Pi(\mu, \nu)$ is the set of probability measures on $(\mathbb{R}^{2d}, \mathcal{B}(\mathbb{R}^{2d}))$ with marginals $\mu$ and $\nu$ respectively. Denote by $\mathcal{P}_2(\mathbb{R}^d)$ the set of all probability measures on $\mathbb{R}^d$ with finite second moment.

## 2 Linear Stochastic Approximation: Setting and Assumptions

Consider the LSA recursion (1) with a deterministic initial point $\theta_0$. We require the following main assumption in this paper:

**A1.** $\{(\mathbf{A}_n, \mathbf{b}_n)\}_{n \in \mathbb{N}^*}$ *is an i.i.d. sequence satisfying the following conditions.*

*(i)* $\mathbb{E}[\mathbf{b}_1] = \bar{b}$ *and there exists* $C_b > 0$ *such that, for any* $u \in \mathbb{S}^{d-1}$, $u^\top(\mathbf{b}_1 - \bar{b}) \in \mathrm{SG}(C_b^2)$.

*(ii)* *There exists* $C_A > 0$ *such that* $\|\mathbf{A}_1\| \leq C_A$ *almost surely.*

*(iii)* *The matrix* $-\bar{A} = -\mathbb{E}[\mathbf{A}_1]$ *is Hurwitz, i.e. for any eigenvalue* $\lambda$ *of* $\bar{A}$, $\mathrm{Re}(\lambda) > 0$.

Both conditions A1-(i), (ii) are standard in analysis of LSA, e.g., in [12, 36, 26]. For example, the assumption on the sub-Gaussianity of $\mathbf{b}_1$ is used in [12] and is relaxed from [36], the almost sure boundedness of $\mathbf{A}_1$ is also used in [12, 36]. Meanwhile, A1-(iii) guarantees the existence of a unique solution $\theta^\star$ to $\bar{A}\theta = \bar{b}$. It is a sufficient and necessary condition for the solution of the ordinary differential equation $\dot{\theta}_t = -\bar{A}\theta_t$ to converge exponentially to $\theta^\star$ [19, Lemma 4.1.2]. The same kind of result holds for the discrete system $\theta_{n+1}^{\mathsf{d}} - \theta_n^{\mathsf{d}} = -\alpha \bar{A}\theta_n^{\mathsf{d}}$.

**Proposition 1.** *Assume that* $-\bar{A}$ *is a Hurwitz matrix. Then there exists a unique positive definite matrix* $Q$ *satisfying the Lyapunov equation* $\bar{A}^\top Q + Q\bar{A} = \mathrm{I}$. *In addition, setting*

$$a = \|Q\|^{-1}/2 \,, \quad and \quad \alpha_\infty = (1/2)\|\bar{A}\|_Q^{-2}\|Q\|^{-1} \,, \tag{3}$$

*then for any* $\alpha \in [0, \alpha_\infty]$, *we get* $\|\mathrm{I} - \alpha\bar{A}\|_Q^2 \leq 1 - a\alpha$. *If in addition* $\alpha \leq \|Q\|^2$ *then* $1 - a\alpha \geq 1/2$.

This result is well known but its proof can be found in Appendix B.1 for completeness. The above proposition implies that the discrete system converges exponentially as $\|\theta_{n+1}^{\mathsf{d}}\| \leq \sqrt{\kappa_Q}(1 - a\alpha)^{n/2}\|\theta_0^{\mathsf{d}}\|$ for $\alpha \in (0, \alpha_\infty)$.

The aim of this paper is to derive high probability bounds on $u^\top\{\theta_n - \theta^\star\}$ for any $n \in \mathbb{N}$, $u \in \mathbb{S}^{d-1}$. Below, we present a counterexample to show that under only A1, if $\alpha > 0$ is fixed, then there exists $\bar{p} > 0$ such that $\lim_{n \to +\infty} \mathbb{E}[\|\theta_n - \theta^\star\|^p] = +\infty$ for $p \geq \bar{p}$. As a corollary, it is impossible to obtain any exponential high probability bounds for $\{\|\theta_n - \theta^\star\| : n \in \mathbb{N}\}$.

**Example 1.** *Consider* (1) *with* $d = 1$ *taking* $\mathbf{b}_n = 0$ *for any* $n \in \mathbb{N}^*$ *and for* $\{\mathbf{A}_n : n \in \mathbb{N}^*\}$ *an i.i.d. sequence of biased Rademacher r.v.s with parameter* $q_A \in (1/2, 1)$:

$$\mathbf{A}_n = \begin{cases} 1 & \text{with probability } q_A \,, \\ -1 & \text{with probability } 1 - q_A \,. \end{cases} \tag{4}$$

*This choice is associated with* $\theta^\star = 0$ *and corresponds to the recursion:* $\theta_n = \prod_{k=1}^n (1 - \alpha\mathbf{A}_k)\theta_0$, *for some* $\theta_0 \neq 0$. *For any* $p \geq 1$ *and* $\alpha \in (0, 1)$, *we have by definition,*

$$\mathbb{E}[|\theta_n|^p] = \{q_A(1-\alpha)^p + (1-q_A)(1+\alpha)^p\}^n |\theta_0|^p \,.$$

*Using the lower bounds* $(1-\alpha)^p \geq 1 - \alpha p$ *and* $(1+\alpha)^p \geq 1 + \alpha p + p(p-1)\alpha^2/2$, *we get for any* $p \geq 1$ *and* $\alpha \in (0, 1)$,

$$\mathbb{E}[|\theta_n|^p] \geq \{1 - p\alpha[(2q_A - 1) - (p-1)\alpha(1-q_A)/2]\}^n |\theta_0|^p \,.$$

*If* $\alpha \in (0, 1)$ *is fixed, then for any* $p > \bar{p}_{q,\alpha} = 1 + 2(2q_A - 1)/[\alpha(1 - q_A)]$, *we have* $\lim_{n \to +\infty} \mathbb{E}[|\theta_n|^p] = +\infty$. *On the other hand, if* $\alpha \in (0, 2(2q_A - 1)/(1 - q_A))$, *then* $\lim_{n \to +\infty} \mathbb{E}[\theta_n^2] = 0$. *Therefore* $\{\theta_n : n \in \mathbb{N}\}$ *converges in distribution to the Dirac measure at* 0 *which corresponds to the unique stationary distribution of this sequence as a Markov chain. In such a case, this distribution admit* $p$ *moments for any* $p \geq 0$.

*However, this result is specific to this particular case and does not hold if only A1 holds. Consider* $\{\theta_n : n \in \mathbb{N}\}$ *defined by* (1) *with* $\{\mathbf{A}_n : n \in \mathbb{N}^*\}$ *given in* (4) *and* $\{\mathbf{b}_n : n \in \mathbb{N}^*\}$ *be an i.i.d. sequence of zero-mean Gaussian random variables with unit variance independent of* $\{\mathbf{A}_n : n \in \mathbb{N}^*\}$. *We show in Appendix B.2 that there exists* $\alpha_{2,\infty}$ *such that for any* $\alpha \in (0, \alpha_{2,\infty}]$, *the Markov chain* $\{\theta_n : n \in \mathbb{N}\}$ *admits a unique invariant distribution* $\pi_\alpha$ *for any* $\alpha > 0$. *Further, for any* $\alpha \in (0, \alpha_{2,\infty}]$ *there exists* $p_\alpha \geq 1$ *such that* $\int_{\mathbb{R}} |\theta|^p \mathrm{d}\pi_\alpha(\theta) = +\infty$ *for any* $p \geq p_\alpha$.

It is, however, possible to obtain any $p$-th moment uniform bound for $\{\|\theta_n - \theta^\star\| : n \in \mathbb{N}\}$ by strengthening A1-(iii) to:

**A2.** *There exist* $\tilde{a} \in (0,1)$, $\tilde{\alpha}_\infty > 0$ *and a positive definite $d$-dimensional matrix $\tilde{Q}$ such that almost surely, for any $\alpha \in (0, \tilde{\alpha}_\infty]$, $\|I - \alpha \mathbf{A}_1\|_{\tilde{Q}} < 1 - \tilde{a}\alpha$.*

Conditions similar to A2 are considered in [9] for the analysis of SA schemes with decreasing stepsize. For example, A2 holds in the case of regularized linear regression. We take $\mathbf{A}_1 = \lambda I + \mathbf{a}_1 \mathbf{a}_1^\top$, for some $\lambda > 0$ and under the assumption that $\|\mathbf{a}_1\|$ is bounded almost surely. The LSA recursion (1) approximates the solution to $(\lambda I + \mathbb{E}[\mathbf{a}_1 \mathbf{a}_1^\top])\theta = \bar{b}$, which admits a unique solution.

On the other hand, examples where A2 does not hold are common. For instance, we may consider TD(0) learning with linear function approximation. For a Markov Reward Process with X as the state space, P : X × $\mathcal{X}$ → [0, 1] as the transition probability, R : X → $\mathbb{R}$ as the reward function, and $\gamma \in (0,1)$ as a discount factor, TD(0) learning is described as in (1) with

$$\mathbf{A}_n = \phi(x_n)\{\phi(x_n) - \gamma\phi(x_n')\}^\top , \quad \mathbf{b}_n = \mathrm{R}(x_n)\phi(x_n) , \tag{5}$$

where $\phi : \mathsf{X} \to \mathbb{R}^d$ is a feature map. A typical setting is when $x_n$ is drawn from the stationary distribution of P and $x_n' \sim \mathrm{P}(x_n, \cdot)$. It is easy to verify A1 provided that $\|\phi(x)\|$, $\mathrm{R}(x)$ are bounded for all $x \in \mathsf{X}$ [38]. However, A2 is violated as $\mathbf{A}_n$ is only rank-one.

Our next endeavor is to establish moment estimates on the product below:

$$\Gamma_{m:n}^{(\alpha)} = \prod_{i=m}^{n}(I - \alpha \mathbf{A}_i) , \quad m, n \in \mathbb{N}^*, \quad m \le n . \tag{6}$$

We also define its expected value as $G_{m:n}^{(\alpha)} = \mathbb{E}[\Gamma_{m:n}^{(\alpha)}] = (I - \alpha \bar{A})^{n-m+1}$. To motivate, we observe that the above product naturally appears after re-centering the LSA recursion (1). For any $n \in \mathbb{N}^*$,

$$\theta_n - \theta^\star = (I - \alpha \mathbf{A}_n)\{\theta_{n-1} - \theta^\star\} + \alpha \varepsilon_n , \quad \varepsilon_n = \mathbf{b}_n - \bar{b} - \{\mathbf{A}_n - \bar{A}\}\theta^\star . \tag{7}$$

An easy induction implies that

$$\theta_n - \theta^\star = \tilde{\theta}_n^{(\mathsf{tr})} + \tilde{\theta}_n^{(\mathsf{fl})} , \quad \tilde{\theta}_n^{(\mathsf{tr})} = \Gamma_{1:n}^{(\alpha)}\{\theta_0 - \theta^\star\} , \quad \tilde{\theta}_n^{(\mathsf{fl})} = \alpha \sum_{j=1}^{n} \Gamma_{j+1:n}^{(\alpha)}\varepsilon_j . \tag{8}$$

The decomposition (8) highlights the two sources of error in the estimation of $\theta^\star$ by $\{\theta_n : n \in \mathbb{N}\}$ which will be separately tackled: $\{\tilde{\theta}_n^{(\mathsf{tr})} : n \in \mathbb{N}\}$ corresponds to the transient (or bias) term and $\{\tilde{\theta}_n^{(\mathsf{fl})} : n \in \mathbb{N}\}$ to the fluctuation term. Both errors are controlled by the product of matrices $\Gamma_{m:n}^{(\alpha)}$, thereby motivating the study of the moment bound on $\Gamma_{1:n}^{(\alpha)}$ as we present next.

## 3 Moment and High-probability Bounds for Products of Random Matrices

Recall from Proposition 1 that the norm of the expected value $G_{1:n}^{(\alpha)} = \mathbb{E}[\Gamma_{1:n}^{(\alpha)}]$ decays exponentially with $n$ as $\|G_{1:n}^{(\alpha)}\| \le \sqrt{\kappa_Q}(1 - \alpha a)^{n/2}$. We expect a similar phenomenon for the moment bound of $\|\Gamma_{1:n}^{(\alpha)}\|$. Precisely, in this section, we show that if $p$ is fixed, then there exists $\alpha_{p,\infty} > 0$ such that for any $\alpha \in (0, \alpha_{p,\infty}]$, the $p$-th moment of $\Gamma_{m:n}^{(\alpha)}$ decays exponentially with $n - m$.

We present the main technical result on the product of general random matrices as follows, whose proof is based on the framework introduced in [18].

**Proposition 2.** *Let $\{\mathbf{Y}_\ell : \ell \in \mathbb{N}\}$ be an independent sequence and $P$ be a positive definite matrix. Assume that for each $\ell \in \mathbb{N}$ there exist $m_\ell \in (0,1)$ and $\sigma_\ell > 0$ such that $\|\mathbb{E}[\mathbf{Y}_\ell]\|_P^2 \le 1 - m_\ell$ and $\|\mathbf{Y}_\ell - \mathbb{E}[\mathbf{Y}_\ell]\|_P \le \sigma_\ell$ almost surely. Define $\mathbf{Z}_n = \prod_{\ell=0}^{n} \mathbf{Y}_\ell = \mathbf{Y}_n \mathbf{Z}_{n-1}$, for $n \ge 1$ and starting from $\mathbf{Z}_0$. Then, for any $2 \le q \le p$ and $n \ge 1$,*

$$\|\mathbf{Z}_n\|_{p,q}^2 \le \kappa_P \prod_{\ell=1}^{n}(1 - m_\ell + (p-1)\sigma_\ell^2)\|P^{1/2}\mathbf{Z}_0 P^{-1/2}\|_{p,q}^2 , \tag{9}$$

*where we recall that $\kappa_P = \lambda_{\mathsf{min}}^{-1}(P)\lambda_{\mathsf{max}}(P)$.*

*Proof of Proposition 2 .* Let $2 \le q \le p$. Consider the following decomposition $\mathbf{Z}_n = \mathbf{Y}_n \mathbf{Z}_{n-1} = (\mathbf{Y}_n - \mathbb{E}[\mathbf{Y}_n])\mathbf{Z}_{n-1} + \mathbb{E}[\mathbf{Y}_n]\mathbf{Z}_{n-1}$, Therefore, we obtain for any $n \in \mathbb{N}$,

$$f_P(\mathbf{Z}_n) = \mathbf{A}_n + \mathbf{B}_n , \quad \mathbf{A}_n = f_P((\mathbf{Y}_n - \mathbb{E}[\mathbf{Y}_n])\mathbf{Z}_{n-1}) , \quad \mathbf{B}_n = f_P(\mathbb{E}[\mathbf{Y}_n])f_P(\mathbf{Z}_{n-1}) ,$$

where $f_P : \mathbb{R}^{d\times d} \to \mathbb{R}^{d\times d}$ is defined for any $B \in \mathbb{R}^{d\times d}$ by $f_P(B) = P^{1/2}BP^{-1/2}$. Since $\mathbb{E}[\mathbf{A}_n|\mathbf{B}_n] = 0$, [18, Proposition 4.3] (see Proposition 10 in Appendix C) implies that

$$\|f_P(\mathbf{Z}_n)\|_{p,q}^2 \leq \|\mathbf{B}_n\|_{p,q}^2 + (p-1)\|\mathbf{A}_n\|_{p,q}^2 . \tag{10}$$

It remains to bound the two terms on the right hand side. To this end, we use [17, Theorem 6.20] which implies that for any $B_1, B_2 \in \mathbb{R}^{d\times d}$,

$$\|B_1 B_2\|_{p,q} \leq \|B_1\| \|B_2\|_{p,q} . \tag{11}$$

As a result and using that for any $B \in \mathbb{R}^{d\times d}$, $\|B\|_P = \|f_P(B)\|$, and $\|\mathbf{Y}_n - \mathbb{E}[\mathbf{Y}_n]\|_P \leq \sigma_n$ we get

$$\|\mathbf{A}_n\|_{p,q} = \left(\mathbb{E}\left[\|f_P(\mathbf{Y}_n - \mathbb{E}[\mathbf{Y}_n])f_P(\mathbf{Z}_{n-1})\|_p^q\right]\right)^{1/q}$$
$$\leq \left(\mathbb{E}\left[\|\mathbf{Y}_n - \mathbb{E}[\mathbf{Y}_n]\|_P^q \|f_P(\mathbf{Z}_{n-1})\|_p^q\right]\right)^{1/q} \leq \sigma_n \|f_P(\mathbf{Z}_{n-1})\|_{p,q} . \tag{12}$$

Similarly, applying $\|\mathbb{E}[\mathbf{Y}_n]\|_P^2 \leq 1 - m_n$

$$\|\mathbf{B}_n\|_{p,q}^2 = \left(\mathbb{E}\left[\|f_P(\mathbb{E}[\mathbf{Y}_n])f_P(\mathbf{Z}_{n-1})\|_p^q\right]\right)^{2/q}$$
$$\leq \left(\mathbb{E}\left[\|\mathbb{E}[\mathbf{Y}_n]\|_P^q \|f_P(\mathbf{Z}_{n-1})\|_p^q\right]\right)^{2/q} \leq (1 - m_n)\|f_P(\mathbf{Z}_{n-1})\|_{p,q}^2 . \tag{13}$$

Combining (12) and (13) in (10) yields for any $n \in \mathbb{N}^*$, $\|f_P(\mathbf{Z}_n)\|_{p,q}^2 \leq (1 - m_n + (p-1)\sigma_n^2)\|f_P(\mathbf{Z}_{n-1})\|_{p,q}^2 \leq \prod_{i=1}^n (1 - m_n + (p-1)\sigma_n^2)\|f_P(\mathbf{Z}_0)\|_{p,q}^2$. The proof is then completed upon using (11) which implies that $\|\mathbf{Z}_n\|_{p,q} = \|P^{-1/2}f_P(\mathbf{Z}_n)P^{1/2}\|_{p,q} \leq \sqrt{\kappa_P}\|f_P(\mathbf{Z}_n)\|_{p,q}$. $\square$

In order to bound $\Gamma_{1:n}^{(\alpha)}$ using Proposition 2, we identify the latter with $\mathbf{Y}_\ell = \mathrm{I} - \alpha\mathbf{A}_\ell, \ell \geq 1, \mathbf{Y}_0 = \mathrm{I}$. As $-\bar{A}$ is Hurwitz, applying Proposition 1 yields $\|\mathbb{E}[\mathbf{Y}_\ell]\|_Q^2 = \|\mathrm{I} - \alpha\bar{A}\|_Q^2 \leq 1 - a\alpha$. Further, A 1-(ii) ensures that almost surely,

$$\|\mathbf{Y}_\ell - \mathbb{E}[\mathbf{Y}_\ell]\|_Q = \alpha\|\mathbf{A}_\ell - \bar{A}\|_Q \leq 2\alpha\sqrt{\kappa_Q}\,\mathrm{C}_A = b_Q\alpha .$$

Therefore, (9) holds with $m_\ell = a\alpha$ and $\sigma_\ell = b_Q\alpha$. As $\|\mathrm{I}\|_p = d^{1/p}$, we obtain the following corollary.

**Corollary 1.** *Assume A 1-(ii)-(iii). Then, for any $\alpha \in [0, \alpha_\infty]$, $2 \leq q \leq p$, and $n \in \mathbb{N}$,*

$$\mathbb{E}^{1/q}\left[\|\Gamma_{1:n}^{(\alpha)}\|^q\right] \leq \|\Gamma_{1:n}^{(\alpha)}\|_{p,q} \leq \sqrt{\kappa_Q}d^{1/p}(1 - a\alpha + (p-1)b_Q^2\alpha^2)^{n/2} , \tag{14}$$

*where $\alpha_\infty$ was defined in* (3)*, and $b_Q = 2\sqrt{\kappa_Q}\,\mathrm{C}_A$.*

Note that Corollary 1 shows $\sup_{n\in\mathbb{N}}\mathbb{E}[\|\Gamma_{1:n}^{(\alpha)}\|^p] < +\infty$ for any $\alpha \in (0, \alpha_{p,\infty}]$, where

$$\alpha_{p,\infty} = \alpha_\infty \wedge a/(2b_Q^2(p-1)) . \tag{15}$$

This kind of condition relating the choice of $\alpha$ with the required order $p$ is necessary as illustrated in Example 1. Corollary 1 further leads to the high-probability bound:

**Corollary 2.** *Assume A 1-(ii)-(iii). Then, for any $\alpha \in (0, \alpha_\infty)$ where $\alpha_\infty$ was defined in* (3)*, $\delta \in (0, 1)$ and $n \in \mathbb{N}$, with probability at least $1 - \delta$,*

$$\|\Gamma_{1:n}^{(\alpha)}\| \leq \sqrt{\kappa_Q}\exp\left[-(an\alpha - \alpha^2 b_Q^2 n)/2 + b_Q\alpha\sqrt{2n\log(d/\delta)}\right] .$$

*Proof.* The result follows from combining Corollary 1 with $p = q$ and Lemma 1 in Appendix C applied with $\mathsf{A} = (-\log(\kappa_Q) + a\alpha n + b_Q^2\alpha^2 n)/2$, $\mathsf{B} = \alpha^2 b_Q^2 n/2$ and $\mathsf{C} = d, p_0 = 2, p_1 = +\infty$. $\square$

The result in Corollary 2 is tight with respect to $\delta$, as illustrated via the following example that continues from Example 1.

**Example** (Continuation of Example 1). *Consider* $\{\theta_n : n \in \mathbb{N}\}$ *defined by* (1) *with* $\{\mathbf{A}_n : n \in \mathbb{N}^*\}$ *given in* (4) *and* $\mathbf{b}_n = 0$ *for any* $n \in \mathbb{N}^*$. *Define*

$$\varphi_q(\alpha) = q_A \log\left(\frac{1+\alpha}{1-\alpha}\right) - \log(1+\alpha), \quad \bar{\alpha}_q = \sup\{\bar{\alpha} > 0 : \varphi_q(\alpha) > 0, \, \forall\, \alpha \in (0,\bar{\alpha})\} \,. \quad (16)$$

*Note that* $\varphi_q(\alpha) \sim \alpha(2q_A - 1)$ *as* $\alpha \downarrow 0$. *Therefore since* $q_A > 1/2$, $\{\bar{\alpha} > 0 : \varphi_q(\alpha) > 0 \text{ for any } \alpha \in (0,\bar{\alpha})\} \neq \emptyset$ *and* $\bar{\alpha}_q$ *is well-defined. Consider also* $\tilde{\varphi}_q(\alpha) = \varphi_q(\alpha) \log^{-1}[(1+\alpha)/(1-\alpha)]$. *Then, we show in Appendix C that for any* $\bar{\delta} \in \left(\mathrm{e}^{-2n\tilde{\varphi}_q(\alpha)}, 1\right)$ *and* $\underline{\delta} \in (\mathrm{e}^{-n\tilde{\varphi}_q^2(\alpha)/(q_A(1-q_A))-2^{-1}\log(n)}, 1)$,

$$\mathbb{P}\left(\theta_n \geq \exp\left(-\varphi_q(\alpha)n + \log\left(\frac{1+\alpha}{1-\alpha}\right)\sqrt{\frac{n\log(1/\bar{\delta})}{2}}\right)\right) \leq \bar{\delta}\,, \quad (17)$$

$$\mathbb{P}\left(\theta_n \geq \exp\left(-\varphi_q(\alpha)n + \log\left(\frac{1+\alpha}{1-\alpha}\right)\sqrt{nq_A(1-q_A)\log(1/\underline{\delta}) + \frac{n\log(n)}{2}}\right)\right) \geq \underline{\delta}\,. \quad (18)$$

*The bounds* (17), (18) *show that the tail distribution associated with* $\theta_n$ *behaves as a log-normal one. If* $\xi$ *follows a zero-mean Gaussian distribution with unit variance, then an easy computation shows that for any* $\sigma > 0$, $\mathbb{P}(\mathrm{e}^{\sigma\xi} \geq t) \sim (2\pi\sigma^2)^{-1/2}\log^{-1}(t)\exp(-(2\sigma^2)^{-1}t^2)$ *as* $t \to \infty$. *Therefore, to have* $\mathbb{P}(\mathrm{e}^{\sigma\xi} \geq t_\delta) \leq \delta$ *for a small* $\delta > 0$, *the scalar* $t_\delta$ *has to be of order* $\exp(\sigma\sqrt{\log(1/\delta)})$.

We conclude the section with a complementary result of Corollary 1 that does not require A1-(ii):

**Proposition 3.** *Assume A1-(iii),* $\|\mathbf{A}_1 - \bar{A}\| \in \mathrm{SG}(\mathrm{C}'_A)$ *for some* $\mathrm{C}'_A > 0$. *Then, for any* $\alpha \in (0, \alpha_\infty)$ *where* $\alpha_\infty$ *was defined in* (3), $2 \leq q \leq p$, *and* $n \in \mathbb{N}$,

$$\mathbb{E}^{1/q}\left[\|\Gamma_{1:n}^{(\alpha)}\|^q\right] \leq \|\Gamma_{1:n}^{(\alpha)}\|_{p,q} \leq \sqrt{\kappa_Q} d^{1/p}(1 - a\alpha + q(p-1)(b'_Q)^2\alpha^2)^{n/2}\,, \quad (19)$$

*where* $b'_Q = 2\sqrt{\kappa_Q}\,\mathrm{C}'_A$.

The proof is similar to that of Proposition 2 and it can be found in Appendix C.

## 4   Finite-time High-probability Bounds for LSA

Relying on the results established in Section 3 and the decomposition (8), we derive high probability bounds on $u^\top\{\theta_n - \theta^\star\}$ for any $n \in \mathbb{N}$ and $u \in \mathbb{S}^{d-1}$, where $\{\theta_n : n \in \mathbb{N}\}$ is defined in (1).

We begin our study with the transient term $\tilde{\theta}_n^{(\mathrm{tr})}$ defined in (8). Observe that

**Proposition 4.** *Assume A1 and let* $p_0 \geq 2$. *Then, for any* $n \in \mathbb{N}^*$, $\alpha \in (0, \alpha_{p_0,\infty})$, *where* $\alpha_{p_0,\infty}$ *is defined in* (15), $u \in \mathbb{S}^{d-1}$ *and* $\delta \in (0,1)$ *it holds with probability at least* $1 - \delta$ *that*

$$|u^\top\Gamma_{1:n}^{(\alpha)}(\theta_0 - \theta^\star)| \leq \sqrt{\kappa_Q} d^{1/p_0}(1 - a\alpha/4)^n\|\theta_0 - \theta^\star\|\delta^{-1/p_0}\,,$$

*where* $a$ *was defined in* (3).

The proof of the above statement is given in Appendix D.1. Proposition 4 only provides a polynomial high probability bound with respect to $\delta$. This is due to the fact that only polynomial moments of $\|\Gamma_{1:n}^{(\alpha)}\|$ up to a maximal order are uniformly bounded in the number of iterations $n$.

We now turn to the fluctuation term $\tilde{\theta}_n^{(\mathrm{fl})}$ defined in (8). Note that under A1, the sequence $\{\varepsilon_n : n \in \mathbb{N}\}$ defined in (7) is i.i.d.. From this observation and following [13], we consider the decomposition

$$\tilde{\theta}_n^{(\mathrm{fl})} = \alpha\sum_{j=1}^{n}\Gamma_{j+1:n}^{(\alpha)}\varepsilon_j = J_n^{(\alpha,0)} + H_n^{(\alpha,0)}\,, \quad (20)$$

where $\{(J_n^{(\alpha,0)}, H_n^{(\alpha,0)}) : n \in \mathbb{N}\}$ are defined by induction for $n \geq 0$ as:

$$\begin{aligned}
J_{n+1}^{(\alpha,0)} &= \left(\mathrm{I} - \alpha\bar{A}\right)J_n^{(\alpha,0)} + \alpha\varepsilon_{n+1}\,, & J_0^{(\alpha,0)} &= 0\,, \\
H_{n+1}^{(\alpha,0)} &= (\mathrm{I} - \alpha\mathbf{A}_n)H_n^{(\alpha,0)} - \alpha(\mathbf{A}_{n+1} - \bar{A})J_n^{(\alpha,0)}\,, & H_0^{(\alpha,0)} &= 0\,.
\end{aligned} \quad (21)$$

The latter recurrence can be written as

$$J_n^{(\alpha,0)} = \alpha \sum_{j=1}^{n} G_{j+1:n}^{(\alpha)} \varepsilon_j , \quad H_n^{(\alpha,0)} = -\alpha \sum_{j=1}^{n} \Gamma_{j+1:n}^{(\alpha)} (\mathbf{A}_j - \bar{A}) J_{j-1}^{(\alpha,0)} .$$

Note that $J_n^{(\alpha,0)}$ is a linear statistics of the random variables $\{\varepsilon_j : j \in \{1, \ldots, n\}\}$ which are centered and i.i.d. under A1. Next, we show that $J_n^{(\alpha,0)}$ is the leading term as the stepsize $\alpha \downarrow 0$. Denote for any $n \in \mathbb{N}^*$ and $\alpha > 0$, the covariance matrix of $J_n^{(\alpha,0)}$ as

$$\mathbf{\Sigma}_n^\alpha = \mathrm{Cov}(J_n^{(\alpha,0)}) . \tag{22}$$

We obtain the following statement which is proven in Appendix D.2:

**Proposition 5.** *Assume A1. Then for any $n \in \mathbb{N}^*$, $\alpha \in (0, \alpha_\infty]$, where $\alpha_\infty$ is defined in (3), $u \in \mathbb{S}^{d-1}$ and $\delta \in (0,1)$, it holds with probability at least $1 - \delta$,*

$$\left| u^\top J_n^{(\alpha,0)} \right| < \mathsf{D}_1 \sqrt{\{u^\top \mathbf{\Sigma}_n^\alpha u\} \log(2/\delta)} + \alpha \sqrt{1 + \log(1/(a\alpha))} \mathsf{D}_2 \log^{3/2}(2/\delta) , \tag{23}$$

*where $\mathsf{D}_1 = 60\sqrt{3}\mathrm{e}^{4/3}$ and $\mathsf{D}_2$ is defined in (49).*

We analyze further the covariance associated with $J_n^{(\alpha,0)}$ and its dependence with respect to $n$ and $\alpha$. First, note that for any $\alpha \in (0, \alpha_{2,\infty}]$, $\{\mathbf{\Sigma}_n^\alpha : n \in \mathbb{N}^*\}$ converges to $\alpha \mathbf{\Sigma}^\alpha$ as $n \to \infty$ where $\mathbf{\Sigma}^\alpha = \alpha \sum_{k=0}^{\infty} G_{1:k} \mathbf{\Sigma}_\varepsilon G_{1:k}^\top$ is the unique solution of the Ricatti equation

$$\bar{A} \mathbf{\Sigma}^\alpha + \mathbf{\Sigma}^\alpha \bar{A}^\top - \alpha \bar{A} \mathbf{\Sigma}^\alpha \bar{A}^\top = \mathbf{\Sigma}_\varepsilon , \quad \text{with} \quad \mathbf{\Sigma}_\varepsilon = \mathbb{E}[\varepsilon_1 \varepsilon_1^\top] . \tag{24}$$

Notice that we focus on the cases where $\mathbf{\Sigma}_\varepsilon$ is full-rank. Using Proposition 1, we obtain that for any $n \geq 0$,

$$\|\mathbf{\Sigma}_n^\alpha - \alpha \mathbf{\Sigma}^\alpha\| \leq \alpha^2 \sum_{k > n} \|G_{1:k}\|^2 \|\mathbf{\Sigma}_\varepsilon\| \leq \alpha a^{-1} \kappa_Q \|\mathbf{\Sigma}_\varepsilon\| (1 - \alpha a)^n . \tag{25}$$

We now give an expansion of $\mathbf{\Sigma}^\alpha$ with respect to $\alpha$. It is well-known that as $\alpha \downarrow 0$, $\mathbf{\Sigma}^\alpha$ converges to $\mathbf{\Sigma}$, the unique solution of the Lyapunov equation (see [34, Lemma 9.1])

$$\bar{A} \mathbf{\Sigma} + \mathbf{\Sigma} \bar{A}^\top = \mathbf{\Sigma}_\varepsilon . \tag{26}$$

Our next result, whose proof is given in Appendix D.3, states the convergence of $\mathbf{\Sigma}^\alpha$ to $\mathbf{\Sigma}$ is of the order of the stepsize $\alpha$.

**Proposition 6.** *Assume that A1-(iii) holds. Then, for any $\alpha \in (0, \alpha_\infty]$, where $\alpha_\infty$ is defined in (3),*

$$\|\mathbf{\Sigma}^\alpha - \mathbf{\Sigma}\|_Q \leq \alpha a^{-1} \|\bar{A} \mathbf{\Sigma} \bar{A}^\top\|_Q ,$$

*where $\mathbf{\Sigma}^\alpha$ and $\mathbf{\Sigma}$ are defined in (24) and (26) respectively and $a$ is given in (3).*

The last step in bounding $\tilde{\theta}_n^{(\mathrm{fl})}$ is to consider $H_n^{(\alpha,0)}$. We proceed similarly to (21) and consider the decomposition $H_n^{(\alpha,0)} = J_n^{(\alpha,1)} + H_n^{(\alpha,1)}$, where $\{(J_n^{(\alpha,1)}, H_n^{(\alpha,1)}) : n \in \mathbb{N}\}$ are defined by induction for $n \geq 0$ as:

$$\begin{aligned} J_{n+1}^{(\alpha,1)} &= (\mathrm{I} - \alpha\bar{A}) J_n^{(\alpha,1)} - \alpha(\mathbf{A}_{n+1} - \bar{A}) J_n^{(\alpha,0)}, & J_0^{(\alpha,1)} &= 0 , \\ H_{n+1}^{(\alpha,1)} &= (\mathrm{I} - \alpha\mathbf{A}_{n+1}) H_n^{(\alpha,1)} - \alpha(\mathbf{A}_{n+1} - \bar{A}) J_n^{(\alpha,1)}, & H_0^{(\alpha,1)} &= 0 . \end{aligned} \tag{27}$$

In our next results, whose proof is given in Appendix D.4, we bound each term of this decomposition separately.

**Proposition 7.** *Assume A1 and let $p_0 \geq 2$. Then, for any $n \in \mathbb{N}$, $\alpha \in (0, \alpha_{p_0,\infty})$, where $\alpha_{p_0,\infty}$ is defined in (15), $u \in \mathbb{S}^{d-1}$ and $\delta \in (0, 1/2)$, with probability at least $1 - 2\delta$, it holds*

$$\left| u^\top J_n^{(\alpha,1)} \right| < \mathrm{e}\mathsf{D}_3 \alpha \log^2(1/\delta) , \quad \left| u^\top H_n^{(\alpha,1)} \right| < \mathsf{D}_4 \alpha p_0^2 \delta^{-1/p_0} , \tag{28}$$

*where $\mathsf{D}_3$ and $\mathsf{D}_4$ are given in (57) and (60), respectively.*

Now we are ready to combine the previous bounds and to state the main result of this section.

**Theorem 1.** *Assume A 1 and let $p_0 \geq 2$. Then, for any $n \in \mathbb{N}$, $\alpha \in (0, \alpha_{p_0,\infty})$, where $\alpha_{p_0,\infty}$ is defined in (15), $u \in \mathbb{S}^{d-1}$ and $\delta \in (0, 1/4)$, with probability at least $1 - 4\delta$, it holds*

$$\alpha^{-1/2}|u^\top(\theta_n - \theta^\star)| < \mathsf{D}_1 \sqrt{\{u^\top \mathbf{\Sigma}^\alpha u\} \log(2/\delta)} + \alpha^{1/2} q^{(1)}(\alpha, \delta) + (1 - a\alpha/4)^n \Delta^{(1)}(\alpha, \delta) , \quad (29)$$

*where $\mathbf{\Sigma}^\alpha$ is the unique solution of (24), $\mathsf{D}_1 = 60\sqrt{3}\mathrm{e}^{4/3}$, $a$ is defined in (3),*

$$
\begin{aligned}
q^{(1)}(\alpha, \delta) &= \left(\mathrm{e}\mathsf{D}_3 \log^2(1/\delta) + \sqrt{1 + \log(1/a\alpha)}\mathsf{D}_2 \log^{3/2}(2/\delta)\right) + \mathsf{D}_4 p_0^2 \delta^{-1/p_0} , \\
\Delta^{(1)}(\alpha, \delta) &= \mathsf{D}_1 \sqrt{a^{-1}\kappa_Q \|\mathbf{\Sigma}_\varepsilon\| \log(2/\delta)} + \sqrt{\kappa_Q} d^{1/p_0} \|\theta_0 - \theta^\star\| \alpha^{-1/2} \delta^{-1/p_0} ,
\end{aligned}
\quad (30)
$$

*where $\kappa_Q$ and $\mathbf{\Sigma}_\varepsilon$ are defined in (3) and (24) respectively.*

*Proof.* The proof follows from the decomposition

$$u^\top(\theta_n - \theta^\star) = u^\top \Gamma_{1:n}^{(\alpha)}(\theta_0 - \theta^\star) + u^\top J_n^{(\alpha,0)} + u^\top J_n^{(\alpha,1)} + u^\top H_n^{(\alpha,1)} ,$$

where $J_n^{(\alpha,0)}$, $J_n^{(\alpha,1)}$ and $H_n^{(\alpha,1)}$ are defined in (21)-(27), the union bound and Proposition 4, Proposition 5, (25) and Proposition 7. □

We now discuss the high probability bound (29). First, the term $\Delta^{(1)}(\alpha, \delta)$, and in particular the initial condition vanishes exponentially fast in the number of iterations $n$. In addition, $q^{(1)}(\alpha, \delta)$ and $\Delta^{(1)}(\alpha, \delta)$ are of order $\delta^{-1/p_0}$ as $\delta \to 0$ and therefore (29) provides polynomial high probability bounds on LSA. However, this conclusion is expected as illustrated in Example 1. Finally, the discussion of (29) with respect to $\alpha$ is postponed to the next section.

Under A2 we can provide a better bound for $H_n^{(\alpha,1)}$.

**Proposition 8.** *Assume A1 and A2. Then, for any $n \in \mathbb{N}$, $\alpha \in (0, \alpha_\infty \wedge \tilde{\alpha}_\infty)$, where $\alpha_\infty$ is defined in (3), $u \in \mathbb{S}^{d-1}$ and $\delta \in (0, 1/2)$, with probability at least $1 - 2\delta$, it holds*

$$\left|u^\top J_n^{(\alpha,1)}\right| < \mathrm{e}\mathsf{D}_3 \alpha \log^2(1/\delta) , \quad \left|u^\top H_n^{(\alpha,1)}\right| < \mathrm{e}\mathsf{D}_5 \alpha \log^2(1/\delta) , \quad (31)$$

*where $\mathsf{D}_3$ and $\mathsf{D}_5$ are given in (57) and (61) respectively.*

As a result, we can establish exponential high probability bounds with respect to $\delta$.

**Theorem 2.** *Assume A 1 and A 2. Then, for any $n \in \mathbb{N}$, $\alpha \in (0, \alpha_\infty \wedge \tilde{\alpha}_\infty)$, $u \in \mathbb{S}^{d-1}$ and $\delta \in (0, 1/4)$, with probability at least $1 - 4\delta$, it holds*

$$\alpha^{-1/2}|u^\top(\theta_n - \theta^\star)| < \mathsf{D}_1 \sqrt{\{u^\top \mathbf{\Sigma}^\alpha u\} \log(2/\delta)} + \alpha^{1/2} q^{(2)}(\alpha, \delta) + (1 - \alpha\tilde{a})^{n/2} \Delta^{(2)}(\alpha, \delta) ,$$

*where $\mathsf{D}_1 = 60\sqrt{3}\mathrm{e}^{4/3}$, $\mathbf{\Sigma}^\alpha$ is solution of (24),*

$$
\begin{aligned}
q^{(2)}(\alpha, \delta) &= \mathrm{e}(\mathsf{D}_3 + \mathsf{D}_5) \log^2(1/\delta) + \sqrt{1 + \log(1/\tilde{a}\alpha)}\mathsf{D}_2 \log^{3/2}(2/\delta) , \\
\Delta^{(2)}(\alpha, \delta) &= \mathsf{D}_1 \sqrt{\tilde{a}^{-1}\kappa_{\tilde{Q}} \|\mathbf{\Sigma}_\varepsilon\| \log(2/\delta)} + \kappa_{\tilde{Q}}^{1/2} \|\theta_0 - \theta^\star\| \alpha^{-1/2} ,
\end{aligned}
\quad (32)
$$

*where $\mathbf{\Sigma}_\varepsilon$ is defined in (24).*

*Proof.* The proof follows the lines of Theorem 1 with Proposition 8 used instead of Proposition 7. □

## 5 Optimality of the derived bounds with respect to $\alpha$: analysis of $(\theta_n)_{n\in\mathbb{N}}$ as a Markov chain

In this section, we study the sequence $\{\theta_n : n \in \mathbb{N}\}$ defined in (1) as a Markov chain. This perspective will allow us to show that the bounds that we derived in Theorem 1 are near-Berstein high probability bounds with respect to the stepsize $\alpha$. Denote by $R_\alpha$ the Markov kernel associated with $\{\theta_n : n \in \mathbb{N}\}$. First, we show that if $\alpha$ is small enough then $R_\alpha$ is geometrically ergodic with respect to the Wasserstein distance of order 2 denoted by $W_2$ and give a representation of its stationary distribution as an infinite sum.

**Theorem 3.** *Assume A1. Then, for any $\alpha \in (0, \alpha_{2,\infty})$, where $\alpha_{2,\infty}$ is defined in (15), $R_\alpha$ admits a unique stationary distribution $\pi_\alpha \in \mathcal{P}_2(\mathbb{R}^d)$ and for any $n \in \mathbb{N}$,*

$$W_2^2(\delta_\theta R_\alpha^n, \pi_\alpha) \leq \kappa_Q d(1 - a\alpha/2)^n \int_{\mathbb{R}^d} \|\tilde{\theta} - \theta\|^2 \mathrm{d}\pi_\alpha(\tilde{\theta}) . \tag{33}$$

*Further, if $\{(\mathbf{A}_k, \mathbf{b}_k) : k \in \mathbb{N}_-\}$ is any sequence of i.i.d. random variables with the same distribution as $(\mathbf{A}_1, \mathbf{b}_1)$, then the following limit exists almost surely and in $\mathrm{L}^2$ and has distribution $\pi_\alpha$:*

$$\theta_\infty^{(\alpha)} = \lim_{n \to -\infty} \theta_n^{(\alpha, \leftarrow)} , \quad \theta_n^{(\alpha, \leftarrow)} = \alpha \sum_{k=n}^{1} \Gamma_{k:0} \mathbf{b}_{k-1} , \quad \Gamma_{k:0} = \prod_{i=k}^{0} (\mathrm{I}_d - \alpha \mathbf{A}_i) . \tag{34}$$

The proof is postponed to Appendix F.1. Based on Theorem 1, we easily get concentration bounds for the family of distributions $\{\pi_\alpha : \alpha \in (0, \alpha_{2,\infty})\}$ around $\theta^\star$.

**Theorem 4.** *Assume A1 and let $p_0 \geq 2$. Then, for any $\alpha \in (0, \alpha_{p_0,\infty})$, where $\alpha_{p_0,\infty}$ is defined in (15), $u \in \mathbb{S}^{d-1}$ and $\delta \in (0, 1/4)$, with probability at least $1 - 4\delta$, it holds*

$$\alpha^{-1/2} |u^\top(\theta_\infty^{(\alpha)} - \theta^\star)| < \mathsf{D}_1 \sqrt{\{u^\top \boldsymbol{\Sigma} u\} \log(2/\delta)} + \alpha^{1/2}[a^{-1/2} \|\bar{A} \boldsymbol{\Sigma} \bar{A}^\top\|_Q^{1/2} + q^{(1)}(\alpha, \delta)] , \tag{35}$$

*where $\boldsymbol{\Sigma}$ is the unique solution of (26), $\mathsf{D}_1 = 60\sqrt{3}\mathrm{e}^{4/3}$, $a$ is defined in (3), and $q^{(1)}(\alpha, \delta)$ in (30).*

*Proof.* The proof follows from Theorem 1, the Portmanteau theorem [22, Theorem 13.16], and the fact that convergence in $W_2$ implies weak convergence. $\qquad\square$

Our results is only polynomial in $\delta$ and we cannot expect improving this dependency as illustrated in Example 1 for fixed $\alpha$. The leading term in (35) as $\alpha \downarrow 0$ is $\sqrt{\mathsf{D}_1\{u^\top \boldsymbol{\Sigma} u\}}$. In our next result, we establish a central limit theorem for the family $(\theta_\infty^{(\alpha)})_{\alpha \in (0, \alpha_{2,\infty}]}$ where $\boldsymbol{\Sigma}$ plays the role of the asymptotic covariance matrix. As a result, (35) is a Bernstein-type high probability bound with respect to $\alpha$ and therefore (35) is sharp. Define for any $\alpha \in (0, \alpha_{2,\infty}]$,

$$\tilde{\theta}_\infty^{(\alpha)} = \alpha^{-1/2}\{\theta_\infty^{(\alpha)} - \theta^\star\} . \tag{36}$$

**Theorem 5.** *Assume A1. Then, the family $\{\tilde{\theta}_\infty^{(\alpha)} : \alpha \in (0, \alpha_{2,\infty}]\}$ converges in law as $\alpha \downarrow 0$ to a zero-mean Gaussian random variable with covariance matrix $\boldsymbol{\Sigma}$ defined by (26).*

Note that this result was established in [30, Theorem 1] for general stochastic approximation schemes but under stronger conditions on the sequence $\{\varepsilon_n : n \in \mathbb{N}^*\}$. In particular, it is assumed that the distribution of $\varepsilon_1$ admits a density with respect to the Lebesgue measure. We relax this condition and provide a new proof for this result. In particular, our strategy to establish Theorem 5 is to consider the decomposition (20) of $\{\theta_n : n \in \mathbb{N}\}$ with $\theta_0 = 0$, since in such case $\theta_n = \tilde{\theta}_n^{(\mathrm{fl})}$ for any $n \in \mathbb{N}$. Define $\{J_n^{(\alpha, \leftarrow)} : n \in \mathbb{N}_-\}$ by

$$J_n^{(\alpha, \leftarrow)} = \alpha \sum_{k=n}^{1} G_{k:0} \varepsilon_{k-1} , \qquad G_{k:0} = \prod_{i=k}^{0}(\mathrm{I} - \alpha \bar{A}) . \tag{37}$$

Note that for any $n \in \mathbb{N}$, $\theta_{-n+1}^{(\alpha, \leftarrow)}$ has the same distribution as $\theta_n^{(\alpha)}$ starting from $\theta_0 = 0$ and $J_n^{(\alpha, 0)}$ as $J_{-n+1}^{(\alpha, \leftarrow)}$. In contrast to $J_n^{(\alpha, 0)}$, $J_{-n+1}^{(\alpha, \leftarrow)}$ admits a limit in $\mathrm{L}^2$ and almost surely denoted by $J_\infty^{(\alpha, \leftarrow)}$. Then, we get for any $u \in \mathbb{S}^{d-1}$, $\alpha \in (0, \alpha_{2,\infty}]$, bounded and Lipschitz function $f : \mathbb{R} \to \mathbb{R}$, with Lipschitz constant smaller than 1, by the Lebesgue dominated convergence theorem

$$|\mathbb{E}[f(u^\top \tilde{\theta}_\infty^{(\alpha, \leftarrow)})] - \mathbb{E}[f(\alpha^{-1/2} u^\top J_\infty^{(\alpha, \leftarrow)})]|$$
$$= \lim_{n \to +\infty} |\mathbb{E}[f(\alpha^{-1/2} u^\top[\theta_{-n+1}^{(\alpha, \leftarrow)} - \theta^\star])] - \mathbb{E}[f(\alpha^{-1/2} u^\top J_{-n+1}^{(\alpha, \leftarrow)})]|$$
$$= \lim_{n \to +\infty} |\mathbb{E}[f(\alpha^{-1/2} u^\top[\theta_n^{(\alpha)} - \theta^\star])] - \mathbb{E}[f(\alpha^{-1/2} u^\top J_n^{(\alpha, 0)})]| \leq \limsup_{n \to +\infty} \mathbb{E}[|\alpha^{-1/2} u^\top H_n^{(\alpha, 0)}|] .$$

Using the decomposition $H_n^{(\alpha, 0)} = J_n^{(\alpha, 1)} + H_n^{(\alpha, 1)}$, where $\{(J_n^{(\alpha, 1)}, H_n^{(\alpha, 1)}) : n \in \mathbb{N}\}$ are defined in (27) and plugging the bounds provided by Proposition 11 and Proposition 12 in Appendix D.4 shows that

$$\limsup_{\alpha \to 0} |\mathbb{E}[f(u^\top \tilde{\theta}_\infty^{(\alpha, \leftarrow)})] - \mathbb{E}[f(\alpha^{-1/2} u^\top J_\infty^{(\alpha, \leftarrow)})]| = 0 .$$

Therefore by the Cramer Wold device and the Portmanteau theorem [22, Theorem 13.16], Theorem 5 follows from the next result.

**Proposition 9.** *Assume A1. Then, for any $u \in \mathbb{S}^{d-1}$, $\{\alpha^{-1/2} u^\top J_\infty^{(\alpha,\leftarrow)} : \alpha \in (0, \alpha_{2,\infty}]\}$ converges in distribution to the zero-mean Gaussian distribution with variance $u^\top \Sigma u$ where $\Sigma$ is given in (26).*

The proof is postponed to Appendix F.2.

## 6    Conclusion

In this paper, we provided a novel non-asymptotic analysis of LSA algorithms with fixed stepsize. For any $\delta \in (0, 1)$, we obtain bounds on the sequence $\{\|\theta_n - \theta^\star\| : n \in \mathbb{N}\}$ that holds with probability at least $1 - \delta$. The bounds are proven to be tight with respect to the stepsize, and we show that such high probability bounds for LSA necessarily have polynomial dependency in $\delta$, leading to a 'heavy-tail' phenomena. Importantly, our results do not require the matrices $\mathbf{A}_n$ to be symmetric but only Hurwitz, which enables one to apply them to various scenarios such as reinforcement learning. Future work includes extending our high probability bounds to a larger panel of random noise, e.g., with heavy tailed distribution, Markovian dependency, as well as Polyak-Ruppert averaging and nonlinear SA.

## Acknowledgement

This work was partly supported by ANR-19-CHIA "SCAI" and CUHK Direct Grant #4055113. The publication was partly supported by the grant for research centers in the field of AI provided by the Analytical Center for the Government of the Russian Federation (ACRF) in accordance with the agreement on the provision of subsidies (identifier of the agreement 000000D730321P5Q0002) and the agreement with HSE University No. 70-2021-00139. Part of this work has been carried out under the auspice of the Lagrange Center in Mathematics and Computing.

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
