# A Summary of Notations and Constants

To facilitate reading the analysis in this paper, the readers can refer to the following table on the variables that are used across the paper for references.

| Variable | Description | Reference |
|---|---|---|
| $\bar{A}$ | $\mathbb{E}[\mathbf{A}_1]$ | A1 |
| $a$ | Real part of minimum eigenvalue of $\bar{A}$ | Proposition 1 |
| $\Gamma_{m:n}^{(\alpha)}$ | Product of random matrices with step size $\alpha$ | (6) |
| $\varepsilon_n$ | Error of LSA after re-centering | (7) |
| $\tilde{\theta}_n^{(\mathrm{tr})}$ | Transient term of LSA error | (8) |
| $\tilde{\theta}_n^{(\mathrm{fl})}$ | Fluctuation term of LSA error | (8) |
| $\alpha_{p,\infty}$ | Bound on $\alpha$ for $\Gamma_{m:n}^{(\alpha)}$ to have bounded $p$-th moment | (15) |
| $J_n^{(\alpha,0)}$ | Dominant term in $\tilde{\theta}_n^{(\mathrm{fl})}$ | (21) |
| $H_n^{(\alpha,0)}$ | Residual term $\tilde{\theta}_n^{(\mathrm{fl})} - J_n^{(\alpha,0)}$ | (21) |
| $\mathbf{\Sigma}^\alpha$ | Limit of covariance matrix of $J_n^{(\alpha,0)}/\sqrt{\alpha}$ | (24) |
| $\mathbf{\Sigma}_\varepsilon$ | Noise covariance $\mathbb{E}[\varepsilon_1 \varepsilon_1^\top]$ | (24) |
| $\mathbf{\Sigma}$ | Limit of $\mathbf{\Sigma}^\alpha$ as $\alpha \downarrow 0$ | Proposition 6 |
| $J_n^{(\alpha,1)}$ | Dominant term in $H_n^{(\alpha,0)}$ | (27) |
| $H_n^{(\alpha,1)}$ | Residual term $H_n^{(\alpha,0)} - J_n^{(\alpha,1)}$ | (27) |

# B Proofs of Section 2

## B.1 Proofs of Proposition 1

The existence and uniqueness of $Q$ follows from [34, Lemma 9.1, p. 140]. Regarding the second statement, note that for any $x \in \mathbb{R}^d \setminus \{0\}$, we have

$$\frac{x^\top (\mathrm{I} - \alpha \bar{A})^\top Q (\mathrm{I} - \alpha \bar{A}) x}{x^\top Q x} = 1 - \alpha \frac{\|x\|^2}{x^\top Q x} + \alpha^2 \frac{x^\top \bar{A}^\top Q \bar{A} x}{x^\top Q x} \ .$$

Hence, we get that for all $\alpha \in [0, \alpha_\infty]$,

$$1 - \alpha \frac{\|x\|^2}{x^\top Q x} + \alpha^2 \frac{x^\top \bar{A}^\top Q \bar{A} x}{x^\top Q x} \le 1 - \alpha \|Q\|^{-1} + \alpha^2 \|\bar{A}\|_Q^2 \le 1 - (1/2)\|Q\|^{-1}\alpha \ .$$

The proof is completed using that for any matrix $\bar{A} \in \mathbb{R}^{d \times d}$, $\|\bar{A}\|_Q \le \kappa_Q^{1/2} \|\bar{A}\|$.

## B.2 Proof for Example 1

The existence and uniqueness of the stationary distribution $\pi_\alpha$ is a consequence of Theorem 3 noting that A1 is satisfied for the particular case that we consider. We now show the second statement. Let $\alpha \in (0, \alpha_{2,\infty})$. First, note that since $\mathbf{b}_1$ is a zero-mean Gaussian random variables with unit variance independent of $\mathbf{A}_1$, we have for any $p \ge 1$,

$$\mathbb{E}[\theta_1^{2p}] = \sum_{k=0}^{2p} \binom{2p}{k} \mathbb{E}[\theta_0^{2p-k}] \mathbb{E}[(1 - \alpha \mathbf{A}_1)^{2p-k}] \mathbb{E}[\mathbf{b}_1^k]$$

$$= \sum_{k=0}^{p} \binom{2p}{2k} \mathbb{E}[\theta_0^{2(p-k)}] \mathbb{E}[(1 - \alpha \mathbf{A}_1)^{2(p-k)}] \mathbb{E}[\mathbf{b}_1^{2k}] \ge \mathbb{E}[\theta_0^{2p}] \mathbb{E}[(1 - \alpha \mathbf{A}_1)^{2p}] \ .$$

This shows that taking $\theta_0$ with distribution $\pi_\alpha$ that if $\int_{\mathbb{R}} |\theta|^{2p} \mathrm{d}\pi_\alpha(\theta) < +\infty$, then it is necessary that $\mathbb{E}[(1 - \alpha \mathbf{A}_1)^{2p}] \le 1$. However, using that $\mathbb{E}[(1 - \alpha \mathbf{A}_1)^{2p}] = \{q_A (1-\alpha)^{2p} + (1 - q_A)(1 + \alpha)^{2p}\}$ and $(1 - \alpha)^{2p} \ge 1 - 2\alpha p$ and $(1 + \alpha)^{2p} \ge 1 + 2\alpha p + 2p(2p - 1)\alpha^2/2$, we get for any $p \ge 1$, $\mathbb{E}[(1 - \alpha \mathbf{A}_1)^{2p}] \ge \{1 - 2p\alpha[(2q_A - 1) - (2p - 1)\alpha(1 - q_A)/2]\}$, therefore $\mathbb{E}[(1 - \alpha \mathbf{A}_1)^{2p}] \le 1$ does not hold for $2p > \bar{p}_{q,\alpha} = 1 + 2(2q_A - 1)/[\alpha(1 - q_A)]$.

## C   Technical and supporting results for Section 3

**Proposition 10** ([18, Proposition 4.3]). *Consider two random matrices* $\mathbf{X}, \mathbf{Y} \in \mathbb{R}^{d \times d}$ *that satisfy* $\mathbb{E}[\mathbf{Y}|\mathbf{X}] = 0$. *Then for* $2 \leq q \leq p$,

$$\|\mathbf{X} + \mathbf{Y}\|_{p,q}^2 \leq \|\mathbf{X}\|_{p,q}^2 + C_p \|\mathbf{Y}\|_{p,q}^2 \ ,$$

*where* $C_p = p - 1$.

**Lemma 1.** *Let* $\mathsf{A} \in \mathbb{R}$, $\mathsf{B} > 0$, $\mathsf{C} \geq 1$, $p_0, p_1 \in \mathbb{R}$ *such that* $1 \leq p_0 \leq p_1 < +\infty$, *and* $X$ *a real random variable satisfying, for any* $p \in [p_0, p_1]$,

$$\mathbb{E}[|X|^p] \leq \mathsf{C} \exp(-\mathsf{A}p + \mathsf{B}p^2) \ . \tag{38}$$

*Then, for all* $\delta \in (0, 1]$, *we have, with probability at least* $1 - \delta$,

$$|X| \leq \exp\left(-\mathsf{A} + \mathsf{B}p_0 + 2\sqrt{\mathsf{B}\log(\mathsf{C}/\delta)} + \log(\mathsf{C}/\delta)/p_1\right) \ , \tag{39}$$

*with the convention* $c/\infty = 0$ *for* $c > 0$. *In addition if* (38) *is satisfied for any* $p \geq p_0$, *then with probability at least* $1 - \delta$,

$$|X| \leq \exp\left(-\mathsf{A} + \mathsf{B}p_0 + 2\sqrt{\mathsf{B}\log(\mathsf{C}/\delta)}\right) \ .$$

*Proof.* Note that by the monotone convergence theorem, it is sufficient to show (38). By Markov's inequality, we have, for any $t > 0$ and $p \in [p_0, p_1]$,

$$\mathbb{P}(|X| \geq t) \leq \mathbb{E}[|X|^p]/t^p \leq \mathsf{C} \exp\left(-p(\log(t) + \mathsf{A} - \mathsf{B}p)\right) \ . \tag{40}$$

Taking $t = \exp(-\mathsf{A} + 2\mathsf{B}a^*)$ for $a^* \in \mathbb{R}$ and maximizing over $p \in [p_0, p_1]$, we obtain

$$\mathbb{P}(|X| \geq \exp(-\mathsf{A} + 2\mathsf{B}a^*)) \leq \mathsf{C} \exp\left(-\mathsf{B}p(2a^* - p)\right) \leq \mathsf{C} \exp\left(-\mathsf{B}\phi(a^*)\right) \ , \tag{41}$$

where

$$\phi(a^*) = \max_{p \in [p_0, p_1]} p(2a^* - p) = (2p_0 a^* - p_0^2)\mathbb{1}_{(-\infty, p_0]}(a^*) + (a^*)^2 \mathbb{1}_{(p_0, p_1)}(a^*) + (2p_1 a^* - p_1^2)\mathbb{1}_{[p_1, +\infty)}(a^*) \ .$$

Note that for any $t \in \mathbb{R}$, the inverse of $\phi$ is given by

$$\phi^{\leftarrow}(t) = \frac{p_0^2 + t}{2p_0}\mathbb{1}_{(-\infty, p_0^2]}(t) + t^{1/2}\mathbb{1}_{(p_0^2, p_1^2)}(t) + \frac{p_1^2 + t}{2p_1}\mathbb{1}_{[p_1^2, +\infty)}(t) \ .$$

For $\delta > 0$, taking $a_\delta^* = \phi^{\leftarrow}(\log(\mathsf{C}/\delta)/\mathsf{B})$ gives

$$\mathbb{P}(|X| \geq \exp[-\mathsf{A} + 2\mathsf{B}\phi^{\leftarrow}(\log(\mathsf{C}/\delta)/\mathsf{B})]) \leq \delta \ .$$

The proof then follows from the fact that for any $t \in \mathbb{R}$, $\phi^{\leftarrow}(t) \leq p_0/2 + \sqrt{t} + t/(2p_1)$. $\square$

**Proof of** (17) **and** (18)

Let $\alpha \in (0, \bar{\alpha}_q)$. Note that by definition of $\{\theta_n : n \in \mathbb{N}\}$ with (4), for any $n \in \mathbb{N}$,

$$\theta_n = (1 - \alpha)^{N_n}(1 + \alpha)^{n - N_n} \ , \quad \text{where } N_n = \sum_{k=1}^n \mathbb{1}_{\{1\}}(Z_k) \ .$$

Then, for any $\beta > 0$, we get

$$\mathbb{P}\left(\theta_n \geq \mathrm{e}^{-\alpha\beta n}\right) = \mathbb{P}\left(\log(\theta_n) \geq -\alpha\beta n\right) = \mathbb{P}\left(N_n \log\left(\frac{1-\alpha}{1+\alpha}\right) \geq -\alpha\beta n - n\log(1+\alpha)\right)$$

$$= \mathbb{P}\left(N_n \leq n \log^{-1}\left(\frac{1+\alpha}{1-\alpha}\right)\{\alpha\beta + \log(1+\alpha)\}\right)$$

$$= \mathbb{P}\left(N_n - q_A n \leq -n\left[q_A - \log^{-1}\left(\frac{1+\alpha}{1-\alpha}\right)\{\alpha\beta + \log(1+\alpha)\}\right]\right) \ . \tag{42}$$

Let $\beta_{\alpha,q} = \alpha^{-1}[q_A \log\{(1+\alpha)/(1-\alpha)\} - \log(1+\alpha)]$. Note that with the condition, $\alpha \in (0, \bar{\alpha}_q)$, $\beta_{\alpha,q} > 0$ and therefore for any $\beta \in (0, \beta_{\alpha,q})$,

$$x_{\alpha,\beta} = \left[q_A - \log^{-1}\left(\frac{1+\alpha}{1-\alpha}\right)\{\alpha\beta + \log(1+\alpha)\}\right] \in (0, \tilde{\varphi}_q(\alpha)) \ , \tag{43}$$

We now show (17). From (42), it follows using Hoeffding's inequality that for any $\beta \in (0, \beta_{\alpha,q})$,

$$\mathbb{P}\left(\theta_n \geq e^{-\alpha\beta n}\right) \leq e^{-2nx_{\alpha,\beta}^2} \ . \tag{44}$$

Hence, for $\bar{\delta} \in (e^{-2n\tilde{\varphi}_q^2(\alpha)}, 1)$, there exists $x \in (0, \tilde{\varphi}_q(\alpha))$ such that $e^{-2nx^2} = \bar{\delta}$ given by $x = \sqrt{\log(1/\bar{\delta})/2n}$, which corresponds by (43) to

$$\beta = \alpha^{-1}\left\{q_A - \sqrt{\log(1/\bar{\delta})/(2n)}\right\}\log\left(\frac{1+\alpha}{1-\alpha}\right) - \alpha^{-1}\log(1+\alpha) \in (0, \beta_{\alpha,q}) \ .$$

This completes the proof of (17) using (44).

We now show (18). Using [1, Lemma 4.7.2] and (42), it holds that for any $\beta \in (0, \beta_{\alpha,q})$,

$$\mathbb{P}\left(\theta_n \geq e^{-\alpha\beta n}\right) \geq \exp(-n\mathrm{KL}(q_A - x_{\alpha,\beta}|q_A) - 2^{-1}\log(n)) \ , \tag{45}$$

where for any $\tilde{q} \in (0, 1)$,

$$\mathrm{KL}(\tilde{q}|q_A) = \tilde{q}\log(\tilde{q}/q_A) + (1-\tilde{q})\log((1-\tilde{q})/(1-q_A)) \ .$$

Note that for any $\tilde{q} \in (0, 1)$, $\tilde{q} \leq q_A$, using $\log(1+t) \leq t$ for any $t > -1$, we get

$$\mathrm{KL}(\tilde{q}|q_A) \leq (q_A - \tilde{q})^2/(q_A(1-q_A)) \ .$$

Therefore, plugging this result into (45) yields for any $\beta \in (0, \beta_{\alpha,q})$,

$$\mathbb{P}\left(\theta_n \geq e^{-\alpha\beta n}\right) \geq \exp(-nx_{\alpha,\beta}^2/(q_A(1-q_A)) - 2^{-1}\log(n)) \ . \tag{46}$$

Hence, for $\underline{\delta} \in (e^{-n\tilde{\varphi}_q^2(\alpha)/(q_A(1-q_A))-2^{-1}\log(n)}, 1)$ there exists $x \in (0, \tilde{\varphi}_q(\alpha))$ such that $e^{-nx^2/(q_A(1-q_A))-2^{-1}\log(n)} = \underline{\delta}$, given by $x = \sqrt{2^{-1}\log(n) + q_A(1-q_A)\log(1/\underline{\delta})/n}$, which corresponds by (43) to

$$\beta = \alpha^{-1}\{q_A - \sqrt{2^{-1}\log(n) + q_A(1-q_A)\log(1/\underline{\delta})/n}\}\log\left(\frac{1+\alpha}{1-\alpha}\right) - \alpha^{-1}\log(1+\alpha) \ .$$

This completes the proof of (18) using (46).

*Proof of Proposition 3.* It suffices to repeat the argument of Corollary 1. We need a version of Proposition 2 for the product $\mathbf{Z}_n = \prod_{\ell=0}^n \mathbf{Y}_\ell$ where $\{\mathbf{Y}_\ell : \ell \in \mathbb{N}\}$ are independent and for each $\ell, q \in \mathbb{N}$ there exist $m_\ell \in (0, 1)$ and $\sigma_{\ell,q} > 0$ such that $\|\mathbb{E}[\mathbf{Y}_\ell]\|_Q^2 \leq 1 - m_\ell$ and $\mathbb{E}^{1/q}[\|\mathbf{Y}_\ell - \mathbb{E}[\mathbf{Y}_\ell]\|_Q^q] \leq \sigma_{\ell,q}$. We use notations of $\mathbf{A}_n, \mathbf{B}_n$ from Proposition 2. Applying independence of $\mathbf{Z}_{n-1}$ and $\mathbf{Y}_n$ and $\mathbb{E}^{1/q}[\|\mathbf{Y}_\ell - \mathbb{E}[\mathbf{Y}_\ell]\|_Q^q] \leq \sigma_{\ell,q}$ we estimate

$$\|\mathbf{A}_n\|_{p,q} \leq \left(\mathbb{E}\left[\|\mathbf{Y}_n - \mathbb{E}[\mathbf{Y}_n]\|_Q^q \|f_Q(\mathbf{Z}_{n-1})\|_p^q\right]\right)^{1/q} \leq \sigma_{n,q}\|f_Q(\mathbf{Z}_{n-1})\|_{p,q}. \tag{47}$$

The bound for $\|\mathbf{B}_n\|_{p,q}^2$ remains the same: $\|\mathbf{B}_n\|_{p,q}^2 \leq (1 - m_n)\|f_Q(\mathbf{Z}_{n-1})\|_{p,q}^2$. Combining this inequality with (47) and (10) yields for any $n \in \mathbb{N}^*$, $\|f_Q(\mathbf{Z}_n)\|_{p,q}^2 \leq (1 - m_n + (p-1)\sigma_{n,q}^2)\|f_Q(\mathbf{Z}_{n-1})\|_{p,q}^2 \leq \prod_{i=1}^n(1 - m_i + (p-1)\sigma_{i,q}^2)\|f_Q(\mathbf{Z}_0)\|_{p,q}^2$. The proof is then completed upon using (11) which implies that $\|\mathbf{Z}_n\|_{p,q} = \|Q^{-1/2}f_Q(\mathbf{Z}_n)Q^{1/2}\|_{p,q} \leq \sqrt{\kappa_Q}\|f_Q(\mathbf{Z}_n)\|_{p,q}$. Finally, it remains to take $\mathbf{Y}_\ell = \mathrm{I} - \alpha\mathbf{A}_\ell, \ell \geq 1$, $\mathbf{Y}_0 = \mathrm{I}$. As $-\bar{A}$ is Hurwitz, applying Proposition 1 yields $\|\mathbb{E}[\mathbf{Y}_\ell]\|_Q^2 = \|\mathrm{I} - \alpha\bar{A}\|_Q^2 \leq 1 - a\alpha$. Further, since $\|\mathbf{A}_\ell - \bar{A}\| \in \mathrm{SG}(\mathrm{C}'_A)$ we get by Lemma 3

$$\mathbb{E}^{1/q}[\|\mathbf{Y}_\ell - \mathbb{E}[\mathbf{Y}_\ell]\|_Q^q] = \alpha\mathbb{E}^{1/q}[\|\mathbf{A}_\ell - \bar{A}\|_Q^q] \leq 2\alpha\sqrt{\kappa_Q q}\,\mathrm{C}'_A = \alpha b'_Q\sqrt{q} \ .$$

Taking $m_\ell = a\alpha$ and $\sigma_{\ell,q} = b'_Q\alpha\sqrt{q}$ we get the claim of the proposition.

$\square$

# D   Proofs of Section 4

For ease of presentation, we drop in this section the dependence of $J_n^{(\alpha,0)}, H_n^{(\alpha,0)}, J_n^{(\alpha,1)}, H_n^{(\alpha,1)}$ with respect to $\alpha$ and simply write $J_n^{(0)}, H_n^{(0)}, J_n^{(1)}, H_n^{(1)}$, respectively. We denote $\tilde{\mathbf{A}}_n = \mathbf{A}_n - \bar{A}$.

## D.1   Proof of Proposition 4

Let $n \in \mathbb{N}^*$, $\alpha \in (0, \alpha_{p_0,\infty}]$, $u \in \mathbb{S}^{d-1}$ and $\delta \in (0,1)$. Under A1, applying Corollary 2 with $p = p_0$ yields

$$\mathbb{E}^{1/p_0}[|u^\top \Gamma_{1:n}^{(\alpha)}(\theta_0 - \theta^\star)|^{p_0}] \leq \mathbb{E}^{1/p_0}[\|\Gamma_{1:n}^{(\alpha)}\|^{p_0}]\|\theta_0 - \theta^\star\|$$
$$\leq \sqrt{\kappa_Q} d^{1/p_0}(1 - a\alpha + (p_0 - 1)b_Q^2 \alpha^2)^{n/2}\|\theta_0 - \theta^\star\| .$$

Since $\alpha \leq a/(2b_Q^2(p_0 - 1))$, using $(1-t)^{1/2} \leq 1 - t/2$ for $t \in [0,1]$, we get

$$\mathbb{E}^{1/p_0}[|u^\top \Gamma_{1:n}^{(\alpha)}\tilde{\theta}_0|^{p_0}] \leq \sqrt{\kappa_Q} d^{1/p_0}\|\theta_0 - \theta^\star\|(1 - a\alpha/4)^n .$$

Applying Markov's inequality easily completes the proof.

## D.2   Proof of Proposition 5

Let $n \in \mathbb{N}^*$, $\alpha \in (0, \alpha_{p_0,\infty}]$, $u \in \mathbb{S}^{d-1}$ and $\delta \in (0,1)$. Using (21) and applying Rosenthal's inequality [32, Theorem 4.1][1] for sum of centered independent random variables we get for any $p \geq 2$,

$$\mathbb{E}[|u^\top J_n^{(0)}|^p] \leq (60\mathrm{e})^p p^{p/2}\{u^\top \Sigma_n^\alpha u\}^{p/2} + \alpha^p 60^p p^p \mathbb{E}\left[\max_{\ell=1,\ldots,n}|u^\top G_{\ell+1:n}\varepsilon_\ell|^p\right] .$$

Applying Lemma 5, we obtain for any $p \geq 2$,

$$\mathbb{E}[|u^\top J_n^{(0)}|^p] \leq (60\mathrm{e})^p p^{p/2}\{u^\top \Sigma_n^\alpha u\}^{p/2} + (9\{1 + \log[1/(a\alpha)]\}\kappa_Q \mathrm{C}_\varepsilon^2)^{p/2}\alpha^p 60^p p^{3p/2} , \quad (48)$$

where the constant $\mathrm{C}_\varepsilon$ is given in (62). Applying Markov's inequality, we get for any $p \geq 2$, $c_1, c_2 > 0$,

$$\mathbb{P}(|u^\top J_n^{(0)}| \geq c_1\{u^\top \Sigma_n^\alpha u\}^{1/2} + c_2)$$

$$\leq \{c_1\{u^\top \Sigma_n^\alpha u\}^{1/2} + c_2\}^{-p}\left[(60\mathrm{e})^p p^{p/2}\{u^\top \Sigma_n^\alpha u\}^{p/2} + (9\{1 + \log[1/(a\alpha)]\}\kappa_Q \mathrm{C}_\varepsilon^2)^{p/2}\alpha^p 60^p p^{3p/2}\right]$$

$$\leq (60\mathrm{e})^p p^{p/2}c_1^{-p} + (9\{1 + \log[1/(a\alpha)]\}\kappa_Q \mathrm{C}_\varepsilon^2)^{p/2}\alpha^p 60^p p^{3p/2}c_2^{-p} .$$

Taking $p = 3\log(2/\delta)$, $c_1 = \mathrm{D}_1(\log(2/\delta))^{1/2}$ and $c_2 = \alpha\sqrt{1 + \log(1/(a\alpha))}\mathrm{D}_2 \log^{3/2}(2/\delta)$ yields the statement, where

$$\mathrm{D}_1 = 60\sqrt{3}\mathrm{e}^{4/3}, \quad \mathrm{D}_2 = 540\sqrt{3}\mathrm{e}^{1/3}\kappa_Q^{1/2}\mathrm{C}_\varepsilon . \quad (49)$$

## D.3   Proof of Proposition 6

**Lemma 2.** *Assume that A1-(iii) holds. Then, for any $\alpha \in (0, \alpha_\infty]$, where $\alpha_\infty$ is defined in (3),*

$$\|\Sigma^\alpha - \Sigma\|_Q \leq \alpha a^{-1}\|\bar{A}\Sigma\bar{A}^\top\|_Q ,$$

*where $\Sigma^\alpha$ and $\Sigma$ are defined in (24) and (26) respectively and $a$ is given in (3).*

*Proof.* Let $\alpha \in (0, \alpha_\infty]$. By definition, (24) and (26) imply

$$\bar{A}(\Sigma^\alpha - \Sigma) + (\Sigma^\alpha - \Sigma)\bar{A}^\top - \alpha\bar{A}(\Sigma^\alpha - \Sigma)\bar{A}^\top = \alpha\bar{A}\Sigma\bar{A}^\top ,$$

which writes

$$\Sigma^\alpha - \Sigma - (I - \alpha\bar{A})(\Sigma^\alpha - \Sigma)(I - \alpha\bar{A})^\top = \alpha^2\bar{A}\Sigma\bar{A}^\top .$$

This implies, by Proposition 1,

$$\|\Sigma^\alpha - \Sigma\|_Q \leq (1 - \alpha a)\|\Sigma^\alpha - \Sigma\|_Q + \alpha^2\|\bar{A}\Sigma\bar{A}^\top\|_Q ,$$

Rearranging terms completes the proof. □

---

[1]Note that the specific universal constants $\mathrm{C}_{\mathrm{R},1} = 60\mathrm{e}$ and $\mathrm{C}_{\mathrm{R},2} = 60$ are not given in the statement, but a close inspection of the proof provide the given estimates.

### D.4 Proof of Proposition 7

Proposition 7 is a direct consequence of the following statements.

**Proposition 11.** *Assume A1. Then, for any $n \in \mathbb{N}$, $\alpha \in (0, \alpha_\infty)$, where $\alpha_{p_0,\infty}$ is defined in (15), $u \in \mathbb{S}^{d-1}$ and $p \geq 2$,*

$$\mathbb{E}[|u^\top J_n^{(1)}|^p] \leq \mathsf{D}_3^p \alpha^p p^{2p} , \tag{50}$$

*where $\mathsf{D}_3$ is given in (57). Moreover, for any $\delta \in (0,1)$ with probability at least $1 - \delta$,*

$$|u^\top J_n^{(1)}| \leq \mathrm{e}\mathsf{D}_3 \alpha \log^2(1/\delta) . \tag{51}$$

**Proposition 12.** *Assume A1 and let $p_0 \geq 2$. Then, for any $n \in \mathbb{N}$, $\alpha \in (0, \alpha_{p_0,\infty})$, where $\alpha_{p_0,\infty}$ is defined in (15), $u \in \mathbb{S}^{d-1}$,*

$$\mathbb{E}[|u^\top H_n^{(1)}|^{p_0}] \leq \mathsf{D}_4^{p_0} \alpha^{p_0} p_0^{2p_0} , \tag{52}$$

*where $\mathsf{D}_4$ is given in (60). Moreover, for any $\delta \in (0,1)$ with probability at least $1 - \delta$,*

$$|u^\top H_n^{(1)}| \leq \mathsf{D}_4 \alpha p_0^2 \delta^{-1/p_0} . \tag{53}$$

*Proof of Proposition 11.* First, we note that (27) implies

$$J_n^{(1)} = \alpha^2 \sum_{\ell=1}^{n-1} S_{\ell+1:n}\varepsilon_\ell, \quad \text{with} \ \ S_{\ell+1:n} = \sum_{k=\ell+1}^{n} (\mathrm{I} - \alpha\bar{A})^{n-k-1}\tilde{\mathbf{A}}_k(\mathrm{I} - \alpha\bar{A})^{k-1-\ell} .$$

It is easy to check that the sequence $(\alpha^2 S_{\ell+1:n}\varepsilon_\ell, \mathfrak{F}_{\ell+1:n})_{\ell=1}^{n-1}$ is a martingale-difference, where $\mathfrak{F}_{\ell+1:n} = \sigma\big((\mathbf{A}_j, \mathbf{b}_j)_{j \in \{\ell+1,\ldots,n\}}\big)$. We may use Burkholders's inequality [16, Theorem 2.10] to get

$$\mathbb{E}[|u^\top J_n^{(1)}|^p] \leq (36p)^p \alpha^{2p} \mathbb{E}\left[\left(\sum_{\ell=1}^{n-1}(u^\top S_{\ell+1:n}\varepsilon_\ell)^2\right)^{p/2}\right] . \tag{54}$$

Using the Minkowski inequality,

$$\mathbb{E}[|u^\top J_n^{(1)}|^p] \leq (36p)^p \alpha^{2p} \left(\sum_{\ell=1}^{n-1} \mathbb{E}^{2/p}[(u^\top S_{\ell+1:n}\varepsilon_\ell)^p]\right)^{p/2} . \tag{55}$$

Denote $V_{\ell+1}^\top = u^\top S_{\ell+1:n}$. Note that by Assumption A1-(ii) and Lemma 1, $\|(\mathrm{I} - \alpha\bar{A})^{n-k-1}\tilde{\mathbf{A}}_k(\mathrm{I} - \alpha\bar{A})^{k-1-\ell}\| \leq \kappa_Q \mathrm{C}_A (1 - \alpha a)^{(n-\ell-2)/2}$. Applying [31, Theorem 3][2], we get for any $t \geq 0$

$$\mathbb{P}\big(\|V_{\ell+1}\| \geq t\big) \leq 2\exp\big\{-t^2/\big(2\kappa_Q^2 \mathrm{C}_A^2(n-\ell)(1 - \alpha a)^{n-\ell-2}\big)\big\} .$$

Using Lemma 3,

$$\mathbb{E}^{2/p}[\|V_{\ell+1}\|^p] \leq 2\sqrt{2}\,\mathrm{C}_A^2\,\kappa_Q^2(n-\ell)(1 - \alpha a)^{(n-\ell-2)}p. \tag{56}$$

Since $S_{\ell+1:n}$ and $\varepsilon_\ell$ are independent,

$$\mathbb{E}[|u^\top S_{\ell+1:n}\varepsilon_\ell|^p] \leq \mathbb{E}[\|V_{\ell+1}\|^p] \sup_{u \in \mathbb{S}^{d-1}} \mathbb{E}[|u^\top \varepsilon_\ell|^p] .$$

Assumption A1-(i), Lemma 3 and Lemma 5 imply, that for any $u \in \mathbb{S}^{d-1}$,

$$\mathbb{E}[|u^\top \varepsilon_\ell|^p] \leq p^{p/2}\,\mathrm{C}_\varepsilon^p(2\sqrt{2})^{p/2} .$$

Combining this bound with (56),

$$\mathbb{E}[|u^\top S_{\ell+1:n}\varepsilon_\ell|^p] \leq p^p(2\sqrt{2})^p\,\mathrm{C}_A^p\,\kappa_Q^p\,\mathrm{C}_\varepsilon^p(n-\ell)^{p/2}(1 - \alpha a)^{(n-\ell-2)p/2} . \tag{57}$$

This inequality and (55) imply

$$\mathbb{E}[|u^\top J_n^{(1)}|^p] \leq p^{2p}\alpha^{2p}(72\sqrt{2})^p\,\mathrm{C}_A^p\,\kappa_Q^p\,\mathrm{C}_\varepsilon^p\left(\sum_{\ell=1}^{n-1}(n-\ell)(1 - \alpha a)^{(n-\ell-2)}\right)^{p/2} \tag{58}$$

$$\leq \alpha^p\mathsf{D}_3^p p^{2p}, \quad \text{where} \quad \mathsf{D}_3 = (72\sqrt{2})\,\mathrm{C}_A\,\kappa_Q\,\mathrm{C}_\varepsilon\,a^{-1}(1 - a\alpha_\infty)^{-1}.$$

---

[2]with $\mathcal{X} = \mathbb{R}^d$ equipped with the Euclidean norm $\|\cdot\|$. Note that $\|x\|, x \in \mathbb{R}^d$ is twice Gateaux differentiable and $\mathcal{X} \in D(A_1, A_2)$ with $A_1 = A_2 = 1$.

Now the equation (28) follows from Markov's inequality. Namely, for any $c_1 > 0$ it holds

$$\mathbb{P}\left(\left|u^\top J_n^{(1)}\right| \geq c_1 \alpha \mathsf{D}_3\right) \leq \frac{\alpha^p \mathsf{D}_3^p p^{2p}}{c_1^p \alpha^p \mathsf{D}_3^p} = c_1^{-p} p^{2p} \,.$$

Taking $p = \log(1/\delta)$ and $c_1 = \mathrm{e} \log^2(1/\delta)$, we obtain (28). $\qquad\square$

*Proof of Proposition 12.* With the decomposition (27), we represent

$$u^\top H_n^{(1)} = -\alpha \sum_{\ell=1}^n u^\top \Gamma_{\ell+1:n}^{(\alpha)} \tilde{\mathbf{A}}_\ell J_{\ell-1}^{(1)} \,.$$

Using Minkowski's inequality,

$$\mathbb{E}^{1/p}\big[\big|u^\top H_n^{(1)}\big|^p\big] \leq \alpha \sum_{\ell=1}^n \mathbb{E}^{1/p}\big[\big|u^\top \Gamma_{\ell+1:n}^{(\alpha)} \tilde{\mathbf{A}}_\ell J_{\ell-1}^{(1)}\big|^p\big] \,.$$

Now, using the independence of $\Gamma_{\ell+1:n}^{(\alpha)}$, $\tilde{\mathbf{A}}_\ell$, $J_{\ell-1}^{(1)}$, and Item (ii),

$$\begin{aligned}
\mathbb{E}^{1/p}\big[\big|u^\top \Gamma_{\ell+1:n}^{(\alpha)} \tilde{\mathbf{A}}_\ell J_{\ell-1}^{(1)}\big|^p\big] &\leq \mathbb{E}^{1/p}\big[\|u^\top \Gamma_{\ell+1:n}^{(\alpha)} \tilde{\mathbf{A}}_\ell\|\big] \sup_{v\in\mathbb{S}^{d-1}} \mathbb{E}^{1/p}\big[\big|v^\top J_{\ell-1}^{(1)}\big|\big] \\
&\leq 2\,\mathsf{C}_A\, \mathbb{E}^{1/p}\big[\|\Gamma_{\ell+1:n}^{(\alpha)}\|^p\big] \sup_{v\in\mathbb{S}^{d-1}} \mathbb{E}^{1/p}\big[\big|v^\top J_{\ell-1}^{(1)}\big|\big] \,.
\end{aligned} \tag{59}$$

Hence, applying Corollary 2 to $\mathbb{E}^{1/p}[\|\Gamma_{\ell+1:n}^{(\alpha)}\|^p]$, and (58) to $\sup_{v\in\mathbb{S}^{d-1}} \mathbb{E}^{1/p}\big[\big|v^\top J_{\ell-1}^{(1)}\big|^p\big]$,

$$\mathbb{E}^{1/p}\big[\big|u^\top H_n^{(1)}\big|^p\big] \leq 2\,\mathsf{C}_A\, \sqrt{\kappa_Q} d^{1/p} \mathsf{D}_3 \alpha^2 p^2 \sum_{\ell=1}^n \big(1 - a\alpha + (p-1)b_Q^2\alpha^2\big)^{(n-\ell)/2} \,.$$

Since $p_0 - 1 \leq a/(2b_Q^2\alpha)$, from the previous estimate it follows

$$\begin{aligned}
\mathbb{E}^{1/p_0}\big[\big|u^\top H_n^{(1)}\big|^{p_0}\big] &\leq 2\,\mathsf{C}_A\, \sqrt{\kappa_Q} d^{1/p_0} \mathsf{D}_3 \alpha^2 p_0^2 \sum_{\ell=1}^n (1-\alpha a)^{(n-\ell)/2} \\
&\leq 4\,\mathsf{C}_A\, \sqrt{\kappa_Q} d^{1/p_0} \mathsf{D}_3 \alpha p_0^2 / a \,.
\end{aligned}$$

Hence,
$$\mathbb{E}\big[\big|u^\top H_n^{(1)}\big|^{p_0}\big] \leq \mathsf{D}_4^{p_0} \alpha^{p_0} p_0^{2p_0} \,, \text{ where } \mathsf{D}_4 = 4\,\mathsf{C}_A\, \sqrt{\kappa_Q} d^{1/p_0} \mathsf{D}_3 / a \,. \tag{60}$$

Using Markov's inequality, we get with probability at least $1 - \delta$,
$$\big|u^\top H_n^{(1)}\big| \leq \mathsf{D}_4 \alpha p_0^2 / \delta^{1/p_0} \,.$$

$\qquad\square$

## D.5 Proof of Proposition 8

The proof is along the same lines as the proof of $H_n^{(1)}$ in Appendix D.4, with the better bound for $\mathbb{E}^{1/p}[\|\Gamma_{\ell+1:n}^{(\alpha)}\|^p]$. For reader's convenience, we provide the proof below. Starting with equation (59), we note that under A2,
$$\mathbb{E}^{1/p}\big[\|\Gamma_{\ell+1:n}^{(\alpha)}\|^p\big] \leq \sqrt{\kappa_{\tilde{Q}}} \big(1 - \alpha\tilde{a}\big)^{n-\ell} \,.$$

We also apply (58) to $\sup_{v\in\mathbb{S}^{d-1}} \mathbb{E}^{1/p}\big[\big|v^\top J_{\ell-1}^{(1)}\big|^p\big]$. Then

$$\mathbb{E}^{1/p}\big[\big|u^\top H_n^{(1)}\big|^p\big] \leq 2\sqrt{\kappa_{\tilde{Q}}}\,\mathsf{C}_A\, \mathsf{D}_3 \alpha^2 p^2 \sum_{\ell=1}^n \big(1 - \alpha\tilde{a}\big)^{n-\ell} \leq \mathsf{D}_5 \alpha p^2 \,,$$

where we have defined

$$\mathsf{D}_5 = 2\sqrt{\kappa_{\tilde{Q}}}\,\mathsf{C}_A\, \mathsf{D}_3 / \tilde{a} \,. \tag{61}$$

Now the equation (31) follows from Markov's inequality.

# E    Concentration results for sub-Gaussian random variables

**Lemma 3.** *Random variable $X \in \mathrm{SG}(\sigma^2)$ for some $\sigma > 0$ if and only if for all $t \geq 0$ the condition $\mathbb{P}(|X| \geq t) \leq 2 \exp\{-t^2/(2\sigma^2)\}$ holds. In addition, in such a case, for any $p \geq 2$, we have*

$$\mathbb{E}[|X|^p] \leq \sqrt{2}\mathrm{e}(2/\mathrm{e})^{p/2}p^{p/2}\sigma^p.$$

*Proof.* The first statement is well-known, see for example [8, Theorem 2.1]. We now show the second statement. By the Fubini theorem, $\mathbb{E}[|X|^p] = p\int_0^{+\infty} u^{p-1}\mathbb{P}(|X| > u)\,\mathrm{d}u$, we get

$$\mathbb{E}[|X|^p] \leq 2p\int_0^{\infty} u^{p-1}\mathrm{e}^{-u^2/(2\sigma^2)}\,\mathrm{d}u = p2^{p/2}\sigma^p\Gamma(p/2),$$

using the change of variable $t = u^2/(2\sigma^2)$. Now, using an upper bound $\Gamma(p/2) \leq (p/2)^{(p-1)/2}\mathrm{e}^{1-p/2}$ (see e.g. [15, Theorem 2]), and $p^{1/2} \leq 2^{p/2}$, we finally get

$$\mathbb{E}[|X|^p] \leq \sqrt{2}\mathrm{e}(2/\mathrm{e})^{p/2}p^{p/2}\sigma^p.$$

$\square$

**Lemma 4.** *Let $\{X_\ell : \ell \in \mathbb{N}\}$ be a sequence of random variables such that $X_\ell \in \mathrm{SG}(\sigma^2)$ for any $\ell \in \mathbb{N}$ and some $\sigma^2 > 0$. Then for any $p \geq 2$,*

$$\mathbb{E}\left[\max_{\ell=1,\ldots,n}\left(|X_\ell|/\sqrt{1+\log\ell}\right)^p\right] \leq 3^p\sigma^p p^{p/2}.$$

*Proof.* Set $a_k = (1 + \log k)^{1/2}$ for $k \in \mathbb{N}^*$. Using the Fubini's theorem, $\mathbb{E}[|\xi|^p] = p\int_0^{+\infty} u^{p-1}\mathbb{P}(|\xi| > u)\mathrm{d}u$, the union bound, and Lemma 3, we get

$$\mathbb{E}\left[\max_{k=1,\ldots,n}\{|X_k|^p/a_k^p\}\right] = p\int_0^{+\infty} u^{p-1}\mathbb{P}\left(\max_{k=1,\ldots,n}|X_k| \geq ua_k\right)\mathrm{d}u$$

$$\leq 2^p\sigma^p + p\int_{2\sigma}^{+\infty} u^{p-1}\mathbb{P}\left(\max_{k=1,\ldots,n}|X_k| \geq ua_k\right)\mathrm{d}u$$

$$\leq 2^p\sigma^p + p\int_{2\sigma}^{+\infty} u^{p-1}\sum_{k=1}^n \mathbb{P}\left(|X_k| \geq ua_k\right)\mathrm{d}u$$

$$\leq 2^p\sigma^p + 2p\int_{2\sigma}^{+\infty} u^{p-1}\sum_{k=1}^n \exp\{-u^2 a_k^2/(2\sigma^2)\}\,\mathrm{d}u$$

$$\leq 2^p\sigma^p + 2p\sigma^p\int_2^{+\infty} y^{p-1}\exp\{-y^2/2\}\Big(\sum_{k=1}^n k^{-y^2/2}\Big)\,\mathrm{d}y$$

$$\leq 2^p\sigma^p + \frac{\pi^2 p\sigma^p}{3}\int_2^{+\infty} y^{p-1}\exp\{-y^2/2\}\,\mathrm{d}y \leq 2^p\sigma^p + \frac{\pi^2 p\sigma^p 2^{p/2-1}}{3}\Gamma(p/2).$$

Using $\Gamma(p/2) \leq (p/2)^{(p-1)/2}\mathrm{e}^{1-p/2}$ (see [15, Theorem 2]), we get

$$\mathbb{E}\left[\max_{k=1,\ldots,n}\{|X_k|^p/a_k^p\}\right] \leq 2^p\sigma^p + \pi^2\sigma^p p^{(p+1)/2}\mathrm{e}^{1-p/2}/(3\sqrt{2}).$$

Since $\sqrt{p}\mathrm{e}^{-p/2} \leq \mathrm{e}^{-1/2}$ and $2^p \leq 4p^{p/2}$,

$$\mathbb{E}\left[\max_{k=1,\ldots,n}\{|X_k|^p/a_k^p\}\right] \leq \sigma^p p^{p/2}\big(4 + \pi^2\mathrm{e}^{1/2}/(3\sqrt{2})\big) < 9\sigma^p p^{p/2}.$$

$\square$

**Lemma 5.** *Assume A1. Then, for any $n \in \mathbb{N}^*$ and $v \in \mathbb{S}^{d-1}$, $v^\top\varepsilon_n$ defined by (7) is a sub-Gaussian random variable with parameter*

$$\mathrm{C}_\varepsilon^2 = 2\,\mathrm{C}_b^2 + 8\,\mathrm{C}_A^2\,\|\theta^\star\|^2. \tag{62}$$

*In addition, for any $n \in \mathbb{N}^*$, $p \geq 2$, $u \in \mathbb{S}^{d-1}$ and $\alpha \in (0, \alpha_\infty)$, it holds*

$$\mathbb{E}\left[\max_{\ell=1,\ldots,n} |u^\top G_{\ell+1:n}^{(\alpha)} \varepsilon_\ell|^p\right] \leq \left(9\kappa_Q p \, \mathrm{C}_\varepsilon^2 \{1 + \log[1/(a\alpha)]\}\right)^{p/2} ,$$

*where $\alpha_\infty$, $a$ and $\kappa_Q$ are defined in (3).*

*Proof.* First we prove (62). Using the representation (7), for any $\lambda \in \mathbb{R}$,

$$\mathbb{E}\big[\exp\big\{\lambda v^\top \varepsilon_n\big\}\big] \leq \mathbb{E}\big[\exp\big\{\lambda v^\top (\mathbf{b}_n - \bar{b} - \{\mathbf{A}_n - \bar{A}\}\theta^\star)\big\}\big]$$
$$\leq \mathbb{E}^{1/2}\big[\exp\big\{2\lambda v^\top (\mathbf{b}_n - \bar{b})\big\}\big]\mathbb{E}^{1/2}\big[\exp\big\{2\lambda v^\top (\bar{A} - \mathbf{A}_n)\theta^\star\big\}\big] .$$

Note that A1-(ii) implies $\left|v^\top (\bar{A} - \mathbf{A}_n)\theta^\star\right| \leq 2\,\mathrm{C}_A \|\theta^\star\|$. Hence, using the Hoeffding inequality, $v^\top (\bar{A} - \mathbf{A}_n)\theta^\star \in \mathrm{SG}(4\,\mathrm{C}_A^2 \|\theta^\star\|^2)$. Combining this result with A1-(i),

$$\mathbb{E}\big[\exp\big\{\lambda v^\top \varepsilon_n\big\}\big] \leq \exp\big\{\lambda^2 \, \mathrm{C}_b^2\big\} \exp\big\{4\lambda^2 \, \mathrm{C}_A^2 \|\theta^\star\|^2\big\} ,$$

yielding the first statement of the lemma.

To prove the second part, let us denote $v_\ell = (\mathrm{I} - \alpha\bar{A})^{n-\ell} u / \|(\mathrm{I} - \alpha\bar{A})^{n-\ell}u\| \in \mathbb{S}^{d-1}$. Using Proposition 1,

$$\mathbb{E}[\max_{\ell=1,\ldots,n} |u^\top G_{\ell+1:n}^{(\alpha)} \varepsilon_\ell|^p] = \mathbb{E}[\max_{\ell=1,\ldots,n} |v_\ell^\top \varepsilon_\ell|^p \|G_{\ell+1:n}^{(\alpha)} u\|^p]$$
$$\leq \kappa_Q^{p/2} \mathbb{E}[\max_{\ell=1,\ldots,n} |v_\ell^\top \varepsilon_\ell|^p (1 - \alpha a)^{p(n-\ell)/2}]$$
$$\leq \kappa_Q^{p/2} \mathbb{E}\left[\max_{\ell=1,\ldots,n} \frac{|v_\ell^\top \varepsilon_\ell|^p}{(1 + \log(n-\ell+1))^{p/2}}\right] \left\{\max_{x>0}(1 + \log(x+1))\mathrm{e}^{-a\alpha x}\right\}^{p/2}$$
$$\leq \kappa_Q^{p/2} (9\,\mathrm{C}_\varepsilon^2 p)^{p/2} \left\{\max_{x>0}[(1 + \log(x+1))\mathrm{e}^{-a\alpha x}]\right\}^{p/2} ,$$

where in the last inequality we used Lemma 4. Set $f(x) = (1 + \log(x+1))\mathrm{e}^{-cx}$ with $c = a\alpha \leq 1$ over $x > 0$. First, note that $f'(x) = \mathrm{e}^{-cx}(1/(1+x) - c - c\log(x+1)) < 0$ for all $x > 1/c - 1$, and thus the maximum is attained for $x \in [0, 1/c - 1]$. Moreover, for any $x \leq 1/c - 1$, we have $f(x) \leq 1 + \log(1 + x) \leq 1 + \log(1/c)$, leading to the desired result. $\square$

# F  Proof of Section 5

## F.1  Proof of Theorem 3

Let $\alpha \in (0, \alpha_{2,\infty})$ and $\lambda_1, \lambda_2 \in \mathcal{P}_2(\mathbb{R}^d)$. By [39, Theorem 4.1], there exists a couple of random variables $\theta_0^{(1)}, \theta_0^{(2)}$ such that $W_2^2(\lambda_1, \lambda_2) = \mathbb{E}[\|\theta_0^{(1)} - \theta_0^{(2)}\|^2]$ independent of $\{(\mathbf{A}_n, \mathbf{b}_n) : n \in \mathbb{N}^*\}$. We introduce then a synchronous coupling between $\lambda_1 R_\alpha^n$ and $\lambda_2 R_\alpha^n$ as follows. Let $\{(\theta_n^{(1)}, \theta_n^{(2)}) : n \in \mathbb{N}\}$ starting from $\theta_0^{(1)}$ and $\theta_0^{(2)}$ respectively and for all $n \geq 0$,

$$\begin{aligned}
\theta_{n+1}^{(1)} &= (\mathrm{I} - \alpha\mathbf{A}_{n+1})\theta_n^{(1)} + \alpha\mathbf{b}_{n+1} \\
\theta_{n+1}^{(2)} &= (\mathrm{I} - \alpha\mathbf{A}_{n+1})\theta_n^{(2)} + \alpha\mathbf{b}_{n+1} .
\end{aligned} \tag{63}$$

Since for all $n \geq 0$, the distribution of $(\theta_n^{(1)}, \theta_n^{(2)})$ belongs to $\Pi(\lambda_1 R_\alpha^n, \lambda_2 R_\alpha^n)$, by definition of the Wasserstein distance we get for any $n \in \mathbb{N}$,

$$W_2(\lambda_1 R_\alpha^n, \lambda_2 R_\alpha^n) \leq \mathbb{E}^{1/2}[\|\theta_n^{(1)} - \theta_n^{(2)}\|^2] = \mathbb{E}^{1/2}[\|\Gamma_{1:n}[\theta_0^{(1)} - \theta_0^{(2)}]\|^2]$$
$$\leq \kappa_Q^{1/2} d^{1/2} (1 - a\alpha/2)^{n/2} W_2(\lambda_1, \lambda_2) , \quad (64)$$

where we have used Corollary 1 for the last inequality. By [39, Theorem 6.16], the space $\mathcal{P}_2(\mathbb{R}^d)$ endowed with $W_2$ is a Polish space. Then, $(\lambda_1 R_\alpha^n)_{n\geq 0}$ is a Cauchy sequence and converges to a limit

$\pi_\alpha^{\lambda_1} \in \mathcal{P}_2(\mathbb{R}^d)$, $\lim_{n \to +\infty} W_2(\lambda_1 R_\alpha^n, \pi_\gamma^{\lambda_1}) = 0$. We show that the limit $\pi_\alpha^{\lambda_1}$ does not depend on $\lambda_1$. Assume that there exists $\pi_\alpha^{\lambda_2}$ such that $\lim_{k \to +\infty} W_2(\lambda_2 R_\alpha^n, \pi_\alpha^{\lambda_2}) = 0$. By the triangle inequality

$$W_2(\pi_\alpha^{\lambda_1}, \pi_\alpha^{\lambda_2}) \le W_2(\pi_\alpha^{\lambda_1}, \lambda_1 R_\alpha^n) + W_2(\lambda_1 R_\alpha^n, \lambda_2 R_\alpha^n) + W_2(\pi_\alpha^{\lambda_2}, \lambda_2 R_\alpha^n) .$$

Thus by (64), taking the limits as $n \to +\infty$, we get $W_2(\pi_\alpha^{\lambda_1}, \pi_\alpha^{\lambda_2}) = 0$ and $\pi_\alpha^{\lambda_1} = \pi_\alpha^{\lambda_2}$. The limit is thus the same for all initial distributions and is denoted by $\pi_\alpha$. Moreover, $\pi_\alpha$ is invariant for $R_\alpha$. Indeed for all $k \in \mathbb{N}^*$, $W_2(\pi_\alpha R_\alpha, \pi_\alpha) \le W_2(\pi_\alpha R_\alpha, \pi_\alpha R_\alpha^n) + W_2(\pi_\alpha R_\alpha^n, \pi_\alpha)$, Using (64) again, we get taking $n \to +\infty$, $W_2(\pi_\alpha R_\alpha, \pi_\alpha) = 0$ and $\pi_\alpha R_\alpha = \pi_\alpha$. The fact that $\pi_\alpha$ is the unique stationary distribution is straightforward by contradiction and using (64). (33) is a simple consequence of (64) taking $\lambda_2 = \pi_\alpha$.

It remains to show that $\theta_\infty^{(\alpha)}$ is well-defined and has distribution $\pi_\alpha$. Since $\{(\mathbf{A}_k, \mathbf{b}_k) : k \in \mathbb{N}_-\}$ is i.i.d., $\sum_{n \le -1} \mathbb{E}^{1/2}[\|\theta_n - \theta_{n+1}\|^2] = \sum_{n \le -1} \mathbb{E}^{1/2}[\|\Gamma_{n:0} \mathbf{b}_{n-1}\|^2] = \sum_{n \le -1} \mathbb{E}^{1/2}[\|\Gamma_{n:0}\|^2] \mathbb{E}^{1/2}[\|\mathbf{b}_{n-1}\|^2]$ and therefore A1-(i) combined with Corollary 1 ensures that this series is finite and therefore $(\theta_n)_{n \in \mathbb{N}_-}$ defined in (34) is a Cauchy sequence almost surely and in $L^2$ which ensures its convergence. Finally, assume now that $\{(\mathbf{A}_k, \mathbf{b}_k) : k \in \mathbb{N}_-\}$ is independent of $\{(\mathbf{A}_k, \mathbf{b}_k) : k \in \mathbb{N}^*\}$. To conclude it is then sufficient to note that if $\theta_0 = \theta_\infty^{(\alpha)}$, then $\theta_1$ has the same distribution as $\theta_\infty^{(\alpha)}$ by definition of the recursion (1).

### F.2 Proof of Proposition 9

Consider a sequence $\{\alpha_n : n \in \mathbb{N}\}$ converging to 0 such that for any $n \in \mathbb{N}$ $\alpha_n \in (0, \alpha_{2,\infty}]$, and let $u \in \mathbb{S}^{d-1}$. For ease of notation, we simply denote $G_{k:0}^{(n)} = G_{k:0}^{(\alpha_n)}$. Note that

$$\alpha_n^{-1/2} u^\top J_\infty^{(\alpha_n, \leftarrow)} = \sum_{k=-\infty}^{1} \Delta M_{n,k} , \qquad \Delta M_{n,k} = \alpha_n^{1/2} u^\top G_{k:0}^{(n)} \varepsilon_{k-1} .$$

By [16, Theorem 3.6], it is sufficient to show that

$$\sup_{k \le 1} \Delta M_{n,k} \xrightarrow[n \to +\infty]{\mathbb{P}} 0 \tag{65}$$

$$\sum_{k \le 1} \Delta M_{n,k}^2 \xrightarrow[n \to +\infty]{\mathbb{P}} u^\top \Sigma u \tag{66}$$

$$\sup_{n \in \mathbb{N}} \mathbb{E}[\sup_{k \le 1} \Delta M_{n,k}^2] < +\infty . \tag{67}$$

First, by Markov inequality and Proposition 1 and A1-(i), we have for any $\eta > 0$ that

$$\mathbb{P}(\sup_{k \le 1} \Delta M_{n,k} \ge \eta) \le \eta^{-4} \mathbb{E}[\sup_{k \le 1} \Delta M_{n,k}^4] \le \eta^{-4} \alpha_n^2 \sum_{k \le 1} \mathbb{E}[\|G_{k:0}^{(n)}\|^4 \|\varepsilon_{k-1}\|^4]$$

$$\le \eta^{-4} \alpha_n^2 \mathbb{E}[\|\varepsilon_0\|^4] \kappa_Q^2 \sum_{k \le 1} (1 - a\alpha_n)^{-2(k-1)} \le \eta^{-4} \alpha_n^2 \mathbb{E}[\|\varepsilon_0\|^4] \kappa_Q^2 (1 - (1 - a\alpha_n)^2)^{-1} ,$$

which shows that (65) holds.

Denote by $\Sigma_n$ the unique solution of the Ricatti equation (24) with $\alpha \leftarrow \alpha_n$. We get by Lemma 2 that there exists $C \ge 0$ such that for any $n \in \mathbb{N}$,

$$\|\Sigma - \Sigma_n\| \le C\alpha_n .$$

Therefore, we obtain that

$$|\sum_{k \le 1} \Delta M_{n,k}^2 - u^\top \Sigma u| \le \alpha_n |\sum_{k \le 1} ((G_{k:0}^{(n)})^\top u)^\top [\varepsilon_{k-1} \varepsilon_{k-1}^\top - \Sigma_\varepsilon] (G_{k:0}^{(n)})^\top u| + C\alpha_n .$$

Then, to establish (66), it remains to show that

$$\alpha_n |\sum_{k \le 1} ((G_{k:0}^{(n)})^\top u)^\top [\varepsilon_{k-1} \varepsilon_{k-1}^\top - \Sigma_\varepsilon] (G_{k:0}^{(n)})^\top u| \xrightarrow[n \to +\infty]{\mathbb{P}} 0 . \tag{68}$$

This follows from A1-(ii)-(i) and Proposition 1 which shows that

$$
\begin{aligned}
\mathbb{E}[|\textstyle\sum_{k\leq 1}&((G^{(n)}_{k:0})^\top u)^\top[\varepsilon_{k-1}\varepsilon_{k-1}^\top - \boldsymbol{\Sigma}_\varepsilon](G^{(n)}_{k:0})^\top u|^2] \\
&= \sum_{k\leq 1}\mathbb{E}\left[|((G^{(n)}_{k:0})^\top u)^\top[\varepsilon_{k-1}\varepsilon_{k-1}^\top - \boldsymbol{\Sigma}_\varepsilon](G^{(n)}_{k:0})^\top u|^2\right] \\
&\leq \sum_{k\leq 1}\|G^{(n)}_{k:0}\|^4\mathbb{E}[\|\varepsilon_{k-1}\varepsilon_{k-1}^\top - \boldsymbol{\Sigma}_\varepsilon\|^2] \leq \mathbb{E}[\|\varepsilon_0\varepsilon_0^\top - \boldsymbol{\Sigma}_\varepsilon\|^2]\kappa_Q^2(1-(1-a\alpha_n)^2)^{-1}\;.
\end{aligned}
$$

Therefore,

$$
\lim_{n\to+\infty}\alpha_n^2\mathbb{E}[|\sum_{k\leq 1}((G^{(n)}_{k:0})^\top u)^\top[\varepsilon_{k-1}\varepsilon_{k-1}^\top - \boldsymbol{\Sigma}_\varepsilon](G^{(n)}_{k:0})^\top u|^2] = 0\;,
$$

which completes the proof of (68).

Finally, we show (67) which follows from A1-(ii)-(i) and Proposition 1,

$$
\begin{aligned}
\mathbb{E}[\sup_{k\leq 1}\Delta M^2_{n,k}] \leq \sum_{k\leq 1}\mathbb{E}[\Delta M^2_{n,k}] &= \alpha_n\sum_{k\leq 1}((G^{(n)}_{k:0})^\top u)^\top\mathbb{E}[\varepsilon_{k-1}\varepsilon_{k-1}^\top](G^{(n)}_{k:0})^\top u \\
&\leq \alpha_n\mathbb{E}[\varepsilon_0\varepsilon_0^\top]\sum_{k\leq 1}\|G^{(n)}_{k:0}\|^2 \leq \mathbb{E}[\varepsilon_0\varepsilon_0^\top]\kappa_Q/a\;.
\end{aligned}
$$