# OpenReview forum: "Tight High Probability Bounds for Linear Stochastic Approximation with Fixed Stepsize"
_NeurIPS.cc/2021/Conference — NeurIPS 2021 Poster_

### Official Review · Reviewer_kUS1 · 2021-06-29

**Rating:** 6
**Confidence:** 3

**Summary:**

The authors analyze the convergence of the linear stochastic approximation (LSA) algorithms with fixed stepsize. They establish a high probability convergence result on how the algorithms estimate compares to the true value of the linear system. They also analyze the stationary distribution of the algorithms iterates. They show that their results are theoretically tight. Their key lemma is a result on bounding the expected value of the norm of products of random matrices, which may be of independent interest.

**Limitations And Societal Impact:**

This was not discussed but I believe it is not needed.

**Main Review:**

The paper seems very interesting mathematically but is difficult to understand in its current form. I believe that the paper needs to be rewritten for clarity.

Pros:

The paper presents tight results on the convergence of the LSA algorithm with minimal assumptions. The techniques used seem interesting and deep, such as the technical lemma used to bound the norm of the mean of a product of random matrices.

Cons:

I believe the clarity of the paper needs to be improve before it is accepted for publication.

The first clarity issue is that the authors do not go into detail on how their results compare and complement previous results on LSA convergence. The authors mention a long list of related papers but simply mention that they are weak. For example, how do the results compare to Theorem 1 of [23]? Can this work provide a high probability version Theorem 1?

The second clarity issue is that to understand the main theorems, the reader needs to understand a large number of symbols defined throughout the text (which are themselves defined with a large number of symbols). The authors should improve the discussion around the main theorems including intuition on the important terms or present simplified theorems to help the reader understand the main claim of the theorems.

Further questions:

Often the mean $\bar{\theta} = \frac{1}{n}\sum_{i=1}^n\theta_i$ is used as the estimate of the true parameters $\theta^*$ due to the natural variation of the iterates. Does this work say anything about the estimate $\bar{\theta}$? Since there is a natural variation of the iterates $\theta_i$ around the true parameters $\theta^*$ the mean should be a better estimate of the true parameters.




**Time Spent Reviewing:**

4

---

> ### Author Response · Authors · 2021-08-10
> **Response to reviewer kUS1**
>
> We thank the reviewer for his/her evaluation of our submission, and now provide answers to the questions and comments:
>
> - Q: clarity
>
>   If our contribution is accepted, we will do our best to clarify the presentation  of our results and better highlight our contributions in relation to the existing literature. Our objective was to present bounds with explicit (and "tight") constants  of the problem. The price to pay is clearly bound expressions that are "complex". To facilitate the reading, we propose  to include a table in the supplement summarizing all the notations we use. In addition, general comments summarizing the main conclusions and consequences of our statements will be made. In particular, we will provide simplified version of our bounds which will highlight the important terms.
>
> - Q: Comparison to previous work on LSA:
>
>   As mentioned in your comment, our main contribution is to make mild "verifiable" assumptions about noise compared to those considered in the existing literature. More precisely, most previous works assume some type of "light tail" conditions (e.g. Assumption 4 in [Lan, 2020]) which mean in the particular case of LSA that for any $n$, $(A_n-A)\theta$ is a sub-Gaussian random variable with a parameter $\sigma^2$ independent of $\theta$. Under this condition, subexponential bounds were reported. Our first contribution is to show that such results cannot be obtained for LSA under general "verifiable" assumptions since only a finite number of moments are uniformly bounded (in the number of iterations). This observation is  already discussed in the paper, but apparently we did not have emphasized this point strongly enough: we plan to expand the discussion of our results  to emphasize the relevance of our work.
>
> - Q: comparison to [23, Theorem 1]:
>
>   Our result cannot be compared  to [23] Theorem 1. [23] Theorem 1 gives a bound on the risk and not a high-probability bound. More precisely, [23] establish a bound on $\mathbb{E}[\| \theta_n - \theta_*\|^2]$ which can be used to derive a HPD but of course the leading term in the deviation will not be subgaussian in  such case in contrast to the result that we obtain. In addition, this paper gives a  deviation inequality  for Polyak-Ruppert averaging procedure: in [23], the successive iterates are averaged whereas we are considering in our contribution only the last iterate. Considering PR averaging is of course a very  interesting question which deserves to be addressed. However, in our opinion, including the same kind of results for the resulting estimators would hinder the main messages and conclusions that are drawn from our analysis.
>
>
> - Q: Polyak-Ruppert averaging:
>
>   Concerning  Polyak-Ruppert averaging of LSA with constant stepsize, results are presented in  [Mou et al, 2020]. However, the approach used in this paper is completely different that the one we take. In  particular, they use the same kind of techniques as [Joulin, Ollivier,2010] and [Frikha, Menozzi, 2012] but  the proof requires the very strong assumption that the gradient noise satisfies a log-Sobolev inequality uniformly in $\theta$.  To our best knowledge, this condition is nearly impossible to verify in practical example, except in very specific ones (i.e. when the gradient can be computed exactly and is perturbed by an additive Gaussian noise). In addition, a careful inspection of the proofs reveal the presence of some gaps which  are most likely difficult to fix.
>
> [Lan, 2020] Lan G. (2020)  "First order and stochastic methods for Machine Learning", Springer Verlag
>
> [Mou et al,2020] Mou, W., Li, C. J., Wainwright, M. J., Bartlett, P. L., & Jordan, M. I. (2020, July). On linear stochastic approximation: Fine-grained Polyak-Ruppert and non-asymptotic concentration. In Conference on Learning Theory (pp. 2947-2997). PMLR.
>
> [Joulin, Ollivier,2010] Joulin, A., & Ollivier, Y. (2010). Curvature, concentration and error estimates for Markov chain Monte Carlo. The Annals of Probability, 38(6), 2418-2442.
>
> [Frikha, Menozzi, 2012] Frikha, N., & Menozzi, S. (2012). Concentration bounds for stochastic approximations. Electronic Communications in Probability, 17, 1-15.

---

> > ### Comment · Reviewer_kUS1 · 2021-08-23
> > **Reply**
> >
> > Thank you for your response. I have updated my score. Please include the clarity improvements in the next version of this work.

---

### Official Review · Reviewer_Sbz9 · 2021-07-13

**Rating:** 6
**Confidence:** 5

**Summary:**

This paper studies the finite-time analysis of linear stochastic approximation algorithms driven by IID noise and fixed stepsize. New analysis based on new results on moments and high probability bounds for products of random matrices. High probability bounds on the LSA iterates are derived under weaker assumptions on the random samples than existing works. The bounds have also been shown tight with the assumptions made and some additional results on the concentration error bounds are also discussed.

**Limitations And Societal Impact:**

More recent works on (non)linear stochastic approximation analysis shall be discussed; and provide also numerical simulations validating the improved error bounds.

**Main Review:**

Finite-sample analysis of stochastic approximation algorithms has been studied in a number of works, recently motivated by e.g., analysis of reinforcement learning algorithms dealing with Markov decision processes. This paper revisits the linear stochastic approximation algorithms and analyzes its finite-time performance using recent results in concentration equalities and moments bounds. Although the analysis is new, and the high probability bound improves upon the existing one, the submission falls in several aspects. First, there are a number of related works analyzing the finite-time error bounds of different stochastic approximation algorithms and, yet they were not discussed in the submission, so it is difficult to judge the novelty and significance of the results. For example,  [a] Gugan Thoppe, Vivek Borkar (2019) A Concentration Bound for Stochastic Approximation via Alekseev’s Formula. Stochastic Systems 9(1):1-26. [b] Borkar, V. S. . "A concentration bound for contractive stochastic approximation." Systems & Control Letters 153.1(2021):104947.
Second, how important is the improvement? Does the analysis carry over to other stochastic approximation algorithms driven by e.g., Markovian noise in reinforcement learning applications? And how about nonlinear stochastic approximation algorithms? Third, it would be great if the authors also provide numerical tests validating the improvement as well as the practical usefulness of the results.

The paper is mostly clear and easy to follow, althoug with a bit heavy notation. Some were used before they were defined.

**Time Spent Reviewing:**

2

---

> ### Author Response · Authors · 2021-08-10
> **Response to reviewer Sbz9**
>
> We thank the reviewer for his/her evaluation of our submission, and now provide answers to the questions and comments:
>
> - Importance of our contributions:
>
>   We agree that there are many works devoted to deriving high probability bounds for stochastic approximation.
>
>  * Most of the results we are aware of assume the so-called "light-tail" conditions, which is equivalent to assume in our case that that $(\mathbf{A}_n - \bar{A}) \theta$ is uniformly in $\theta$ sub-gaussian, which of course cannot be true (see for example Assumption 4, in [Lan, 2020]). Under this assumption, which fails to be true for LSA (but also in all situations where the gradients is not uniformly bounded), "usual" subexponential deviation inequalities are derived. The first originality of our result is to establish that in the case of LSA with a constant step, one cannot expect to obtain "usual" deviation inequalities. In particular, we establish that there is only a finite number of bounded moments. We show through example 1 that this is not an artifact of the proof but the "true" behavior of the algorithm. We provide a tight condition for the number of bounded moments. We thus show the perhaps surprising fact that, even in situations where the gradient noise is sub-Gaussian, the estimators $\{\theta_n\}$ have only a finite number of bounded moments and are thus "heavy-tailed". The heavy-tailness does not come here from "heavy-tailness" of the gradient, but rather from the multiplicative structure of the update. Note that the situation is markedly different if instead of using a fixed stepsize, we would have used a sequence of decreasing stepsize.
>
>  * The second originality of our approach is to obtain "precise" high probability deviation inequalities in the sense that the leading term in the bound corresponds exactly to the Gaussian deviation of the algorithm (the bound we give is "asymptotically" exact). We thus have a "Gaussian" regime (which is the leading term when the learning step is "vanishingly small") and a "heavy-tail" regime when the learning step is large.
>
> - Q: Comparison with previous works and nonlinear SA:
>
>   * [a] Gugan Thoppe, Vivek Borkar (2019) A Concentration Bound for Stochastic Approximation via Alekseev’s Formula. Stochastic Systems 9(1):1-26. deals with decreasing step sizes algorithms (and not constant learning rate). It is not strictly speaking a "finite-time" bound, since the bound is valid  only "at convergence" (for all iteration indices larger than $n_0$ where $n_0$ is some function of $\epsilon$, see Theorem 1). The objective of [a] is different from ours. It rather seeks to establish the asymptotic behavior of the algorithm in the neighborhood of a locally stable attractor (LASE) which is not necessary assumed to be unique. \
>   More precisely, given a  precision $\epsilon > 0$ and that the current iterate is in a neighbourhood of a LASE, [a] provides an estimate for i.) the time required to hit the $\epsilon$−ball of this LASE, and ii.) the probability that after this time the iterates are indeed within this $\epsilon$−ball and stay there thereafter. The latter estimate can also be viewed as the ‘lock-in’ probability. The result is very different. It is a local result, i.e., it gives a bound on the probability of convergence to a LASE if the iterates land up in its domain of attraction eventually. Also, the results are only "partially" quantitative (of course, some of the constants showing up in theorems can be made completely explicit, but this would require to dig into the proofs). The assumptions are of course different.  Since the results in [a] apply to a more general context but provides much weaker conclusions, the assumptions used in [a] are weaker. The results we propose are of a very different nature: they are non-asymptotic and give precise deviation inequalities.
>    * [b] "A concentration bound for contractive stochastic approximation" provides high-probability bounds for contractive stochastic approximation. The paper deals with decreasing stepsize (whereas we are considering constant step sizes). This paper also assumes that each iteration is coercive (with the notation of [b], $F: \mathbb{R}^{d} \mapsto \mathbb{R}^{d}$  and $F_{n}(x):=F(x)+M_{n+1}(x)$  satisfy, for some $\alpha \in(0,1)$ and  $K \in(0, \infty)$,
>     ** Cond-1) $\|F(x)-F(y)\| \leq \alpha\|x-y\|$, $x, y \in \mathbb{R}^{d}$ and
>     ** Cond-2) $\left\|F_{n}(x)\right\| \leq K+\alpha\|x\|$ a.s. \
>  None of these assumptions are satisfied in linear SA.  For the first assumption (Cond-1) to be satisfied we would have to suppose that the spectral norm of the matrix $A$ is strictly less than 1. For the second condition to be satisfied (cond-2), we would have to suppose that the spectral norm of the random matrices $\{\mathbf{A}_n\}$ are also strictly less than 1 (which in particular implies that these matrices are of full rank;  we of course does not use such assumption which most often fail to be true in the applications that we consider).
>
>   Thus, although there are several finite time bounds in stochastic approximation, we have not found any that cover linear stochastic approximation under "verifiable" assumptions in the applications we wish to study (regression, value function approximation, etc...).
>
>   We have also not found any work that highlights the very particular type of concentration we get for LSA with a constant stepsize and that is related to the fact that there are only a finite number of moments that do not diverge exponentially.
>
>   [Lan, 2020] "First order and stochastic methods for Machine Learning", Springer Verlag
>
> - Q: Markov samples
>
>   We notice that deriving (tight) high probability bounds for LSA beyond the iid $\{ A_n, b_n \}$ case is difficult. We found that it is not trivial to extend the current proof to the Markovian case. However, as noted in Example 1 which shows the impossibility of an exponential high probability bound in the iid case, we notice that the latter will also be impossible for the Martingale setting.
>
>   We are unaware of any works in the literature that analyzes the tight high probability bounds for LSA without the iid sample assumption. One of our future directions is to investigate the extension of our analysis to cases such as Markovian samples.
>
> - Q: Numerical experiments
>
>   If our contribution is accepted, we will include a numerical example illustrating the impossibility result in Example 1.

---

> > ### Comment · Reviewer_Sbz9 · 2021-09-22
> > **Updated score**
> >
> > The authors have address most of my concerns. And based on my read of other reviews, I have updated score. The authors are suggested to have a comprehensive review of recent works on finite-sample analysis stochastic approximation schemes (including TD and Q-learning algorithms), to highlight the added value and importance of the results and/or analysis presented in this paper broadly. Some closely related ones include e.g.,  [1] A concentration bound for contractive stochastic approximation." Systems & Control Letters 153.1(2021):104947. [2] A Concentration Bound for Stochastic Approximation via Alekseev’s Formula. Stochastic Systems, 9(1):1-26, 2019. [3] A multistep Lyapunov approach for finite-time analysis of biased stochastic approximation." arXiv:1909.04299, 2019. [4] Finite-sample analysis of nonlinear stochastic approximation with applications in reinforcement learning, arXiv:1905.11425, 2019. [5] On linear stochastic approximation: Fine-grained Polyak-Ruppert and non-asymptotic concentration. COLT, 2020.

---

### Official Review · Reviewer_eosb · 2021-07-15

**Rating:** 7
**Confidence:** 3

**Summary:**

This paper provides high probability concentration bounds for linear stochastic approximation algorithms using constant stepsize under weaker assumptions. The key novelty is to derive tight moment bounds for the product of random matrices. The polynomial concentration bounds were shown to be tight. Since linear SA was widely used in application, the results are of broader interest to the NeurIPS community.

**Limitations And Societal Impact:**

No Societal Impact

**Main Review:**

Although this paper is theoretical and highly technical, it is well-organized to convey the main idea.

(line 25): $Q$-learning cannot be modeled by a linear stochastic approximation algorithm since it involves a max operator.

Often in applications such as reinforcement learning, the samples $\{(A_n,b_n)\}$ are not i.i.d.. For example, in TD-learning with synchronous update, the corresponding samples form a martingale difference sequence. In TD-learning with asynchronous update,  the corresponding samples form a Markov chain. It may be an interesting future direction to extend the high probability bounds in this paper to these more general settings.



**Time Spent Reviewing:**

2

---

> ### Author Response · Authors · 2021-08-10
> **Response to reviewer eosb**
>
> We thank the reviewer for his/her positive evaluation of our submission, and now provide answers to the questions and comments:
>
> - Q: Q-learning
>
>   The reviewer is right. The Q-learning algorithm belongs to a nonlinear stochastic approximation scheme. We will remove this reference in the final version.
>
> - Q: Markov samples
>
>   Thanks for raising this interesting question. We notice that deriving (tight) high probability bounds for LSA beyond the iid $\{ A_n, b_n \}$ case is difficult. We found that it is not trivial to extend the current proof to the Markovian case. However, as noted in Example 1 which shows the impossibility of an exponential high probability bound in the iid case, we notice that the latter will also be impossible for the Martingale setting.
>
>   We are unaware of any works in the literature that analyzes the tight high probability bounds for LSA without the iid sample assumption. One of our future directions is to investigate the extension of our analysis to cases such as Markovian samples.

---

> > ### Comment · Reviewer_eosb · 2021-08-21
> > **Comparison to related literature**
> >
> > Thank the authors for their feedback. I read the author feedback as well as other reviewers' comments. Some reviewers raised concerns about comparison with related work. However, since the results offered in this paper are of different forms compared to existing literature, it is not clear how to conduct such comparison. That being said, it would be better if the high probability bounds offered in this paper are of the form $P(\vert|\theta_n-\theta^*\vert|>\cdot)$. Is it possible to translate the results of this paper to the above described form? If it is possible, then one can use the union bound to translate the results of this paper to compare with [Thoppe and Borkar 2019][Borkar 2021].
> >
> > There are some works that study high probability bounds of contractive stochastic approximation algorithms with Markovian noise (such as [Li et al, 2020][[Qu and Wierman 2020]). Although they are not specifically for linear SA, it is very likely that the analysis can be extended to study linear SA algorithms.
> >
> > [Gugan Thoppe, Vivek Borkar (2019) A Concentration Bound for Stochastic Approximation via Alekseev’s Formula. Stochastic Systems 9(1):1-26.] \
> > [Borkar, V. S. . "A concentration bound for contractive stochastic approximation." Systems & Control Letters 153.1(2021):104947]
> > [Li, G., Wei, Y., Chi, Y., Gu, Y., & Chen, Y. (2020). Sample complexity of asynchronous Q-learning: Sharper analysis and variance reduction. arXiv preprint arXiv:2006.03041.]
> > [Qu, G., & Wierman, A. (2020, July). Finite-Time Analysis of Asynchronous Stochastic Approximation and $ Q $-Learning. In Conference on Learning Theory (pp. 3185-3205). PMLR.]

---

> > > ### Author Response · Authors · 2021-08-22
> > > **comparison with related works**
> > >
> > > We have been very specific about the originality of our results compared to previous work, and we have stressed in the paper and in our answers that those  do not apply to  the linear stochastic approximation. We know that this may seem shocking as it is clear that LSA is of course the most basic example of stochastic approximation and that it is difficult to find such a universally used algorithm, but this is the truth and that is why our result is important. All the previous reported works on HPD for SA give conditions allowing to obtain "classical" Bernstein-type concentration inequalities. In this paper, we highlight that for LSA, we cannot obtain a "classical" Bernstein inequality, and this  is  why our results in not a mere "extension" or "adaptation" of the  previous works! We prove that there are two regimes, a "Gaussian" regime for "small" values of the step and a "polynomial" regime that reflects the fact that the stochastic LSA approximation has only a finite number of moments that are uniformly bounded in time. This reflects the existence of a non-sub-Gaussian regime for large values of the step, which is not present at all in the previously known bounds. We show that this non-sub-Gaussian regime is not a by-product of our proof, but is characteristic of the problem we study (roughly speaking, the originality of this concentration inequality comes from the very special nature of the concentration of matrix products).
> > >
> > > The reason why LSA does not admit a usual sub-Gaussian regime is that the assumption that is usually made to obtain HPD in SA is a uniform sub-Gaussian deviation inequality between the random field and its mean-field approximation (previous results with a few exceptions study gradient algorithms, this is not the case for our results). This hypothesis is not verified in LSA and that is why the concentration is not usual. Our work uses very recent results on the concentration of products of random matrices (which we improve in passing). It is therefore quite incorrect to say that our work "extends" existing results. They are of a different nature, we highlight deviation behaviors unknown in the literature of SA

---

> > > > ### Author Response · Authors · 2021-08-22
> > > > **Comments on Thoppe Borkar**
> > > >
> > > > First, this paper deals with decreasing step size, while our work  deals with constant step size. Concentration inequalities with decreasing step size is different from that with constant step size, even in the LSA case; This comes from the fact that all p-th moments of the estimates remain asymptotically bounded in LSA with decreasing step size, which is not the case with constant step size. Our results are "non-local", they remain valid for all initial conditions while the results in Thope and Borkar are "local" (see for example their theorem 1.1). The type of concentration obtain in Thope and Borkar is "classical", which is not surprising under the assumptions formulated in this paper. The connexion between our results and Thope and Borkar is therefore quite tenuous! The assumptions are very different, and the conclusions are also different.

---

> > > > ### Author Response · Authors · 2021-08-22
> > > > **Comments on Borkar 2021**
> > > >
> > > > We already have answered this question in another comment. Of course, the paper by V. Borkar "A concentration bound for contractive stochastic approximation" is very valuable. It provides high-probability bounds for contractive stochastic approximation. The paper deals with decreasing stepsize whereas we are considering constant step sizes. As we have stressed, concentration properties for constant and decreasing stepsize are different. This paper also assumes that each iteration is coercive (with the notation of this paper,
> > > > Cond-1 $ \|F(x)-F(y)\| \leq \alpha\|x-y\|, \quad x, y \in \mathscr{R}^{d}$
> > > > Cond-2 $ \left\|\widetilde{F}_{n}(x)\right\| \leq K+\alpha\|x\| \quad \text { a.s. }$
> > > >
> > > > The conditions we use are strictly weaker. For the first assumption (Cond-1) to be satisfied we would have to suppose that the spectral norm of the matrix  is strictly less than 1. We simply assume that -A is Hurwitz, which is in line with the stability of deterministic linear systems.
> > > > For the second condition to be satisfied (cond-2), we would have to suppose that the spectral norm of the random matrices
> > > >  are also strictly less than 1 (which in particular implies that these matrices are of full rank; we of course do not use such assumption which most often fail to be true in the applications that we consider - the update are most of the time of rank one, so (5) in Borkar cannot be satisfied).
> > > >
> > > > Thus, although there are several finite time bounds in stochastic approximation, we have not found any that cover linear stochastic approximation under "verifiable" assumptions in the applications we wish to study (regression, value function approximation, etc...).
> > > >
> > > > We have also not found any work that highlights the very particular type of concentration we get for LSA with a constant stepsize and that is related to the fact that there are only a finite number of moments that do not diverge exponentially.
> > > >
> > > > [Lan, 2020] "First order and stochastic methods for Machine Learning", Springer Verlag

---

> > > > ### Author Response · Authors · 2021-08-22
> > > > **Comments on "Asynchronous Q-learning on a single Markovian trajectory"**
> > > >
> > > > This paper is quite a departure from what is being written, as it focuses very specifically on Q-learning and not on LSA. "Given the Markovian trajectory
> > > > $\{s_t, a_t, r_t\}$
> > > > generated by the behavior policy $\pi_b$, the asynchronous Q-learning algorithm maintains a Q-function estimate $Q_t : S × A → \mathbb{R}$ at each time t and adopts the following iterative update rule
> > > > $$Q_{t}\(s_{t-1}, a_{t-1})=(1-\eta_{t}) Q_{t-1}(s_{t-1}, a_{t-1})+\eta_{t} T_{t}(Q_{t-1})(s_{t-1}, a_{t-1})$$
> > > > and
> > > > $$ Q_{t}(s, a)=Q_{t-1}(s, a), \quad \forall(s, a) \neq\left(s_{t-1}, a_{t-1}\right)$$
> > > > where $T_t$ denotes the empirical Bellman operator'
> > > > This is "really" a paper devoted to $Q$-learning, see their Theorem 1. This is a very nice paper, but very far from our main scope of interest

---

> > > > ### Author Response · Authors · 2021-08-22
> > > > **Comments on "Finite-Time Analysis of Asynchronous Stochastic Approximation and Q-Learning"**
> > > >
> > > > The paper considers an *asynchronous* SA scheme: at each iteration, only a single component of the parameter vector is updated. More precisely, for $i= i_t$,
> > > > $$
> > > > x_{i}(t+1)=x_{i}(t)+\alpha_{t}\left(F_{i}(x(t))-x_{i}(t)+w(t)\right) \quad \text { for } i=i_{t},
> > > > $$
> > > > and for $i \ne i_t$
> > > > $$
> > > > x_{i}(t+1)=x_{i}(t)
> > > > $$
> > > > The first Assumption  is that $F$ is a contraction  in a weighted infinity norm
> > > > $$
> > > > \|F(x)-F(y)\|_{v} \leq \gamma\|x-y\|_{v}
> > > > $$
> > > > where the weighted infinity norm is defined in Definition 1 of this paper. \
> > > > The second condition is that the $w(t)$ is a bounded martingale difference.
> > > >
> > > > The third assumption is that the stepsize is
> > > > $$
> > > > \alpha_t = \frac{h}{t+t_0}
> > > > $$
> > > >
> > > > The fourth assumption is that, a.s.
> > > > $$
> > > > \| x(t) \|_v \leq \bar{x}
> > > > $$
> > > >
> > > > 1) We consider synchronous stochastic approximation (asynchronous gives rise to different problems)
> > > > 2) We consider fixed learning rate: the concentration inequality for fixed learning rate is markedly different than decreasing learning rate.
> > > > 3) We do not assume that the "noise" is uniformly bounded (this is not the case for LSA)
> > > > 4) We do not assume a $\gamma$-contraction but rather assume that $-A$ is Hurwitz which corresponds to the classical stability condition for linear systems.
> > > > 5) We derive a Berstein-type bound - our result is of course globally very different
> > > >
> > > > The assumptions and context of this paper are "completely" different from ours. So are the results. It is really difficult to connect the two results!

---

### Official Review · Reviewer_jZ9d · 2021-07-16

**Rating:** 6
**Confidence:** 3

**Summary:**

The paper studies the problem of analyzing linear stochastic approximation (LSA) algorithms which are used to obtain an approximate solution of a linear system $ A \theta = b$, where $A$ and $b$ are observed through random variables $\{ (A_1,b_``1), (A_2,b_2),\cdots \}$. The analysis relies on new results regarding moments and high probability bounds for matrix products under weak conditions with polynomial concentration with the order depending on the step-size $\alpha$, which diverges from results in prior work. At this level of generality, the authors show that polynomial tails cannot be improved in the worst case. The concentration bounds are in terms of covariance matrices appearing in central limit theorems of the matrix products under appropriate scaling limits.

**Ethical Concerns:**

None, to the best of my knowledge

**Limitations And Societal Impact:**

Yes, I believe the authors have adequately addressed the limitations of their work, and future work to be done.

**Main Review:**

The paper considers the classical problem of analyzing LSA algorithms which have gained renewed interest in modern applications such as least squares (and ridge) regression, reinforcement learning (such as TD(0) learning) among other areas. Convergence results are established for these algorithms with a fixed stepsize under a set of assumptions more general than those considered in prior work. Indeed to the best of my knowledge, this is the first paper to establish convergence properties with non-subgaussian tails under a wide family of assumptions. I think this paper is a fine contribution to the theory of analysis of LSA algorithms. However, I think it would help the interpretability of the results if more instantiations of these results are provided for relevant applications (ridge regression / TD(0) learning, etc.).

- Under assumption A.1, it would help to include a remark as to how concretely show how this assumption compares with that in prior works.

- Line 120: $\mathcal{X}$ notation is incorrect. Line 134: The terminology suggesting that the matrix $G^{(\alpha)}_{1:n}$ decays with $n$ is mathematically imprecise.

- Concerning the discussion on TD(0) learning under linear function approximation: is there any intuition how the current results can be extended to the case when $x_n$ is not sampled from the stationary distribution but equals $x_{n-1}'$. More generally, when the i.i.d.ness assumptions fail, but a martingale structure is retained, is there any intuition as to whether such bounds still hold true? For the purpose of the exposition (and comparison), it may help to instantiate the proved results in the context of TD(0) learning to compare with known results here.

**Time Spent Reviewing:**

5

---

> ### Author Response · Authors · 2021-08-10
> **Response to reviewer jZ9d**
>
> We thank the reviewer for his/her positive evaluation of our submission, and now provide answers to the questions and comments:
>
> - Q: instantiations of the results:
>
>   Thanks for the suggestion. If space is allowed, in the final version we will provide examples of applying Theorem 1 on instantiations of LSA schemes such as ridge regression and TD(0) learning.
>
> - Q: Assumption A1:
>
>   The assumption A1 is standard in the analysis of LSA. We will expand the justifications about the assumption in the final version.
>
>   For example, A1-(i) on the sub-Gaussianity of ${\bf b}_n$ is used in [11] and is relaxed from [34], A1-(ii) is used in [11, 34], and A1-(iii) on the Hurwitzness of $-\bar{A}$ is assumed in all the previous literature. Note that A1-(iii) is a necessary condition for the ODE $\dot{\theta}_t = -\bar{A}\theta_t$ to converge to $\theta^\star$. This is a necessary condition for the algorithm to be convergent. If there is an eigenvalue whose real part is strictly positive, then the algorithm diverges exponentially. Finally, making use of Proposition 3 of our paper, A1-(ii) could be relaxed and primarily computation suggests that high probability bounds could be also derived but at the price of more involved constants. Besides, the bounds would be different from the ones we obtained. That is why we have decided to postponed this study to future work.
>
> - Q: exponential convergence of $G_{1:n}^{\alpha}$
>
>    Proposition 1 implies that $E[ || G_{1:n}^{\alpha} || ] \leq \sqrt{\kappa_Q} ( 1 - a \alpha )^{n/2}$, i.e., it decays exponentially with $n$. We will make this statement more precise in the final version.
>
> - Q: Martingale structure:
>
> Thanks for raising this interesting question. We notice that deriving (tight) high probability bounds for LSA beyond the iid $\{ A_n, b_n \}$ case is of course possible but is even more difficult than the already non trivial i.i.d. case. The first key to the proof is to establish the exponential stability of the product of matrices. The proof is "relatively" simple in the i.i.d. case using the ideas of [17]. It is possible to extend this proof in the Markovian case (uniformly geometrically ergodic in particular) and in the case of stationary dependent processes (beta-mixing for example), but the proof is then much more delicate. We can then prove the validity of the decomposition used in Section 4. But here again, we have to work differently. For Proposition 5, it is necessary to obtain a Bernstein bound, which is still possible in the Markovian case. However, as noted in Example 1 which shows the impossibility of an exponential high probability bound in the iid case, we notice that the latter will also be impossible for the Martingale setting.
>
>   We are unaware of any works in the literature that analyzes the tight high probability bounds for LSA without the iid sample assumption. One of our future directions is to investigate the extension of our analysis to cases such as Markovian samples (which includes Martingale samples as a special case).

---

### Official Review · Reviewer_kP8y · 2021-07-28

**Rating:** 7
**Confidence:** 3

**Summary:**

The paper provides a non-asymptotic analysis for linear stochastic approximation algorithms. Polynomial concentration bounds with order depending on the stepsize are derived under weaker conditions than previous works. The paper also shows that logarithmic dependence on probability $\delta$ is not possible.

**Limitations And Societal Impact:**

The authors have adequately addressed the limitations and potential negative societal impact of their work

**Main Review:**

The main contribution of the paper is:

1. Under mild conditions on the sequence of $A_n, b_n$, the paper provides non-asymptotic bound for the approximation error with polynomial dependence on $1/\delta$ and near optimal dependence on the step size.

2. Counterexamples are provided illustrating that the logarithmic dependence in $1/\delta$ cannot hold.

3. The optimality of the term $\sqrt{\alpha}$ is shown via a central limit theorem for the family of stationary distribution for $\{\theta_n\}.

Overall, I believe this paper is a nice complement to the literature of linear stochastic approximation. My questions are summarized as below:

1. Example 1 is concluded with the impossibility to obtain exponential high probability bounds for $||\theta_n-\theta^*||$. Does this naturally suggest the impossibility to obtain exponential high probability bounds for $u^\top(\theta_n-\theta^*)$ for some fixed $u$? This seems correct to me. But it would be great to make it explicit in the paper.

2. Following-up the first point, is it possible to derive reasonable upper bounds for the term $||\theta_n-\theta^*||$ instead of $u^\top(\theta_n-\theta^*)$? Simple union bound seems to introduce exponential dimension dependence. I'm curious whether this is the best one can do or not.

3. Is the asymptotic covariance matrix $\Sigma$ in Equation (2) guaranteed to be full rank and well-conditioned? If not, in the degenerate case when $u^\top \Sigma u$ is approximately 0, is the dependence w.r.t $\alpha$ still tight?

**Update based on the author response:

Thanks for the detailed reply. I'm satisfied with the responses to the comments and remain my score.


**Time Spent Reviewing:**

5

---

> ### Author Response · Authors · 2021-08-10
> **Response to reviewer kP8y**
>
> We thank the reviewer for his/her positive evaluation of our submission, and now provide answers to the questions and comments:
>
> - Q: Impossibility of exponential bound for $u^\top(\theta_n - \theta^*)$:
>
>   The reviewer is correct saying that it is also impossible to obtain exponential high probability bound for $u^\top ( \theta_n - \theta^* )$ (for fixed $u$). Notice that Example 1 is analyzed for the scalar problem with $d=1$. In the final version, we will add a comment on that.
>
> - Q: High probability bound for $||\theta_n - \theta^*||$:
>
>   This is an interesting question to study the high probability upper bound for the term $||\theta_n - \theta^*||$. Instead of introducing a simple union bound (perhaps with an $\epsilon$-net argument), a crude estimate can also be obtained as follows. We first substitute $u = e_i$ in Theorem 1 for all $i=1,...,d$, which leads to
>
>   $$ \mathbb{P}( \alpha^{-1/2} | [ \theta_n - \theta^\star ]_i | \geq C(\alpha,n,\delta) ) \leq 4 \delta $$
>
>   implying (by union bound)
>
>   $$ \mathbb{P}( \alpha^{-1/2} || \theta_n - \theta^\star ||_{\infty} \geq C(\alpha,n,\delta) ) \leq 4 d \delta $$
>
>   By using the  equivalence norm $||x||_\infty \geq ||x||_2 / \sqrt{d}$, we therefore get polynomial bounds with respect to $d$.
>
> - Q: degenerate covariance matrix:
>
>   Thank you for this interesting question. The asymptotic covariance matrix is defined in (27), depending on the covariance $\Sigma^{\epsilon} = E[ \epsilon_1 \epsilon_1^\top ]$, where $\epsilon_1$ is the noise vector defined in (7). Notice that under A1, the matrix $A$ must be Hurwitz and is thus full rank and well conditioned. Consequently, if $\Sigma^\epsilon$ is also full-rank and well conditioned, the matrix $\Sigma$ will also be full rank and well conditioned.
>
>   The degenerate case mentioned by the reviewer pertains to a situation when the noise covariance $\Sigma^\epsilon$ is degenerate. In this case, the reviewer is correct that $u^\top \Sigma u$ can be small for some $u$. We didn't provide a direction dependent bound in our theorem for simplicity. However, we note that it is easy to extend the analysis to derive a direction dependent bound in Lemma 5.

---

### Official Review · Reviewer_9FBX · 2021-07-29

**Rating:** 7
**Confidence:** 1

**Summary:**

This paper performs a non-asymptotic analysis for the linear stochastic approximation (LSA) algorithm, with bounds given in the form of Eq. (2). This paper shows the tightness of their bounds by examples.

**Limitations And Societal Impact:**

### Limitations

1. The paper claims tightness by showing a CLT-like statement. However, this seems only justifying the variance term in the main result (see, e.g., the $\sqrt{u^\top \Sigma u}$ term in Eq. 2), but not the constant term (e.g., $\delta^{-\frac{1}{p_0}}$ in Eq. 2), while the latter term is of more interest.


### Societal Impact

The paper is mainly theoretical, and I don't see any potential negative societal impact.


**Main Review:**

The LSA algorithm is significant in machine learning. I am generally unfamiliar with the related work, but I think this paper makes a good technical contribution to the study of LSA.

The proof is written formally, but the intuition is unclear to me.

**Time Spent Reviewing:**

6

---

> ### Author Response · Authors · 2021-08-10
> **Response to reviewer 9FBX**
>
> We thank the reviewer for his/her positive evaluation of our work, and now provide answers to his questions and comments:
>
> - Q: Intuition behind our proof:
>
>   Our proofs rely on the concentration inequality for products of matrices provided in Section 3. The main intuition is that that a "heavy-tail" phenomena appears as soon as the iteration of LSA is, with positive probability, not contractive (i.e. A2 is not verified), as the error term is in essence multiplicative, and can thus reach exponentially large values with non-negligible probability.
>
> - Q: CLT-like argument:
>
>   In Section 5, optimality is considered w.r.t. the behavior when $\alpha$ tends to $0$. The polynomial concentration term in $1/\delta$ is thus considered negligible in this regime. Example 1 shows that the bound is at least in $C_1 + \sqrt{\alpha} C_2 + O(\alpha)$, while Theorem 5 shows that the leading term $C_1$ is at least proportional to $\sqrt{u^\top\Sigma u}$ (i.e. the leading term of tight concentration inequalities for zero-mean Gaussian random variables with covariance matrix $\Sigma$). We will improve this discussion in the final version. Moreover, we would like to note that finding the exact power of the polynomial term is probably a very difficult problem, and is left for future work.

---

### Official Review · Reviewer_Pfgj · 2021-07-30

**Rating:** 6
**Confidence:** 2

**Summary:**

The paper offers a non-asymptotic analysis of the linear stochastic approximation algorithm with fixed step-size, where the solution to the linear system $\bar{A} \theta = \bar{b}$ is approximated iteratively. While prior literature has considered asymptotic and non-asymptotic bounds, the main contribution of this work is the improved quality of non-asymptotic bound for LSA performance, yielding more meaningful finite-time confidence sets. The authors consider a set of mild assumptions on the coefficients, under which they prove high probability bounds on the approximation error. The analysis is based on a new analysis of product of random matrices, with further extension to random matrices with Hurwitz mean but not necessarily symmetric a.s. The authors further discuss the tightness of the resulted polynomial bound with respect to error probability $\delta$ through a clean (counter-)example (Example 1), and the sharpness with respect to the step-size through derivation of a central limit theorem.

**Limitations And Societal Impact:**

The theoretical limitations are discussed. N/A for negative societal impact.

**Main Review:**

The paper is well-written and organized, with clear and consistent notations. The structure of having a standalone section (Sec 3) on product bound of matrices followed by its application to LSA (Sec 4) and discussion of tightness (Sec 4-5) seems natural and easy to follow. A minor suggestion is to be more explicit when discussing tightness (e.g. L60-61: “Section 5 shows the tightness of the bounds”) since more than one variables are considered.

The concentration results proven in the paper are nontrivial and clearly improve upon previously known analysis (asymptotic or non-asymptotic) in the sense that the assumptions are weak and that the bound itself is tight. Though, I would hope for a more substantial discussion of the statements (L32-34) “non-asymptotic studies are in general too coarse and lose significant statistical information in their derivation”, especially when claiming “their upper bounds are generally loose when used in predicting the actual performance of LSA.”

While the proof technique isn’t a most novel one mathematically in my opinion, the details are sound and well-presented and I believe the main claims are all correct. The main technical tool relates Huang et al’s recent paper on matrix concentration for products. The idea of handling the error term by splitting it into bias and variance terms is nice albeit standard and the actual proofs are clearly nontrivial to carry out.

### Other comments/suggestions:

I like the recurrent use of Example 1 to demonstrate tightness of the polynomial bound, a very clean example. This could be a trivial point, but I would appreciate some intuition for Assumption A2 as the additional condition for exponential high probability bounds. In particular, as is illustrated by the few examples in Section 2, A2 holds in many natural cases but not others, and I would love to have a conceptual understanding of which cases may admit an exponential bound vs polynomial.

In the discussion of tightness in Section 5, I didn’t completely get how the paragraph L271-275 discussing Eqn (36) on the limit $\theta_\infty^\alpha$ implies the optimality of the finite-time bound (or does it not and is simply a standalone result?). I could be missing something obvious, but perhaps some additional interpretations around Theorem 5 and (36) can be helpful.

I would also love to see a discussion of implications and take-away message of the results, advancing it further from a mostly mathematical result. E.g., what are some insights behind the necessity of Assumption 2? How can the results improve current practice/knowledge when applied to reinforcement learning for example?

### Note:
I may not be familiar enough with some pieces of related work. I have checked some of the proofs carefully but not all.

### Typos spotted:
L69-70: probabilities measure -> probability measures

L113: definite positive -> positive definite

L120: \mathcal{X} -> \mathsf{X}

L268: in $\mathsf{D}_1$ definition, $e^{2/3}$ -> $e^{4/3}$ I believe, in consistency with other occurrences

L286: functions -> function

L490: are an independent -> are independent

L498: remains take -> remains to take

L520: first term RHS in the equation below: missing an $\alpha$ subscript

L556: first “=” in equation below should be \le

**Time Spent Reviewing:**

20

---

> ### Author Response · Authors · 2021-08-10
> **Response to reviewer Pfgj**
>
> We thank the reviewer for his/her thorough evaluation of our paper and positive comments. Below is a list of answers to the several comments and questions of this review:
>
> - Q: "tightness" in Section 5 and optimality of our bound:
>
>   "Tightness" refers to the behavior in $\alpha$: we show that the leading term of our bounds with respect to $\alpha$ as this parameter goes to $0$ corresponds (up to universal constants) to the variance in the central limit theorem that we establish.  We will make this more visible in the final version.
>
> - Q: discussion of prior works (L32-34):
>
>   Most prior works assume a uniform sub-Gaussian noise on the "gradient" step, i.e., that $(A_{n+1} - A) \theta_{n}$ is $\sigma$-sub-Gaussian where $\sigma$ is independent of $\theta_n$; see ("[Lan, 2020] - p.116 Assumption 4"). Under general condition on $(A_n)_n$, it is expected that this condition does not hold for LSA.
>
> - Q: Assumption A2:
>
>   Intuitively, assumption A2 means that one iteration of LSA is an almost sure contraction, hence leading to exponential convergence with high probability. This situation arises when the noise terms are almost surely negligible compared to the deterministic terms. For example, if the matrix $A_n$ is symmetric and its smallest eigenvalue is almost surely larger than some $\varepsilon > 0$, then A2 is satisfied. However, note that A2 fails in many cases of interest, e.g., for linear systems, in which case the bound is polynomial.
>
> - Q: Takeaway message:
>
>   The takeaway message is that a "heavy-tail" phenomena appears as soon as the iteration of LSA is, with positive probability, not contractive (i.e. A2 is not verified). This implies that sub-Gaussian concentration is not possible for LSA, and extremely large values for the iterates can occur. In practice, our results advocate for practices such as gradient clipping, as iterates of TD with a fixed stepsize will sometimes explode, resulting in instabilities and hazardous behavior of the overall ML pipeline.
>
> - Finally, we thank the reviewer for all the typos found in the document and will update it accordingly.
>
> [Lan, 2020] Lan G. (2020)  "First order and stochastic methods for Machine Learning", Springer Verlag

---

### Decision · Program_Chairs · 2021-09-27

**Decision:**

Accept (Poster)

**Comment:**

The paper provides a non-asymptotic analysis of linear stochastic approximation algorithms. It shows interesting concentration bounds and certain bounds are not achievable. The majority of the reviewers agree that the analysis is interesting, and provides new results to the literature. I have read the paper carefully and also the discussions between the authors and the reviewers, and agree with the majority of the reviewers. One reviewer pointed out that its connections to other papers are not made clear and pointed out some future directions to work on, and the authors have responded to these particular points. I do agree that the current form of the paper has not addressed fully these interesting new directions, but it has crossed the threshold.